

# Global carbonyl sulfide (OCS) measured by MIPAS/Envisat during 2002–2012

Norbert Glatthor[1], Michael Höpfner[1], Adrian Leyser[1], Gabriele P. Stiller[1], Thomas von Clarmann[1], Udo Grabowski[1], Sylvia Kellmann[1], Andrea Linden[1], Björn-Martin Sinnhuber[1], Gisele Krysztofiak[2], and Kaley A. Walker[3]

[1]Karlsruher Institut für Technologie, Institut für Meteorologie und Klimaforschung, Karlsruhe, Germany
[2]University of Orléans, LPC2E, CNRS, Orléans, France
[3]Department of Physics, University of Toronto, Toronto, Canada

*Correspondence to:* Norbert Glatthor (Norbert.Glatthor@kit.edu)

**Abstract.** We present a global OCS data set covering the period June 2002 to April 2012, derived from FTIR limb emission spectra measured with the Michelson Interferometer for Passive Atmospheric Sounding (MIPAS) on the ENVISAT satellite. The vertical resolution is 4–5 km in the height region 6–15 km and 15 km at 40 km altitude. The total estimated error amounts to 40–50 pptv between 10 and 20 km and to 120 pptv at 40 km altitude. MIPAS OCS data show no systematic bias with

respect to balloon observations, with deviations mostly below ±50 pptv. However, they are systematically higher than the OCS volume mixing ratios of the ACE-FTS instrument on SCISAT, with maximum deviations of up to 100 pptv in the altitude region 13–16 km. The data set of MIPAS OCS exhibits only moderate interannual variations and low interhemispheric differences. Average concentrations at 10 km altitude range from 480 pptv at high latitudes to 500–510 pptv in the tropics and at northern mid-latitudes. Seasonal variations at 10 km altitude amount up to 35 pptv in the northern and up to 15 pptv in the southern

hemisphere. Northern hemispheric OCS abundances at 10 km altitude peak in June in the tropics and around October at high latitudes, while the respective southern hemispheric maxima were observed in July and in November. Global OCS distributions at 250 hPa (∼10–11 km) show enhanced values at low latitudes, peaking during boreal summer above the western Pacific and the Indian Ocean, which indicates oceanic release. Further, a region of depleted OCS amounts extending from Brazil to central and southern Africa was detected at this altitude, which is most pronounced in austral summer. This depletion is

related to seasonally varying vegetative uptake by the tropical forests. Typical signatures of biomass burning like the southern hemispheric biomass burning plume are not visible in MIPAS data, indicating that this process is only a minor source of tropospheric OCS. At the 150 hPa level (∼13–14 km) enhanced amounts of OCS were also observed inside the Asian Monsoon Anticyclone, but this enhancement is not especially outstanding as compared to other low latitude regions at the same altitude. At the 80 hPa level (∼17–18 km) equatorward transport of mid-latitude air masses containing lower OCS amounts around the

summertime anticyclones was observed. A significant trend could not be detected in tropospheric MIPAS OCS amounts, which points to globally balanced sources and sinks.



## 1 Introduction

First measurements of atmospheric carbonyl sulfide (OCS), which is the most prevalent sulfur-containing trace gas, were performed by Hanst et al. (1975) and Sandalls and Penkett (1977). Its relative long tropospheric lifetime of 3–4 years (Chin and Davis, 1993; Griffith et al., 1998) favours its enrichment in the troposphere and enables its ascent into the stratosphere,

where during volcanically unperturbed periods it acts as dominant source of sulfur as prerequisite of the stratospheric aerosol layer (Crutzen, 1976; Kremser et al., 2016). Since stratospheric aerosols have a significant contribution to climate forcing, good knowledge of sources, sinks and fluxes of atmospheric sulfur is of scientific interest. Anthropogenic sources of atmospheric sulfur can be quantified rather accurately, but the characteristics of the natural sulfur cycle are still not sufficiently well known. One reason is the lack of widespread measurements of OCS to better constrain our knowledge on its sources and sinks. In this

respect the comprehensive data set of MIPAS can give further insight on the global OCS distribution.

Various scientific studies have been performed to quantify natural and anthropogenic sulfur emissions and sinks (Eriksson, 1960; Junge, 1963; Robinson and Robbins, 1968; Kellogg et al., 1972; Friend, 1973; Adams et al., 1981; Andreae, 1990; Chin and Davis, 1993; Kuhn et al., 1999). These authors reported strongly differing values, reflecting large uncertainties in determination of sulfur fluxes. More recent estimates of the OCS budget were given by Watts (2000) and Kettle et al. (2002).

In these estimates the oceans are generally assumed to be the dominant source and vegetative uptake to be the dominant sink.

While in earlier estimates the OCS budget often was unbalanced, sources and sinks are better balanced in Kettle et al. (2002). The direct flux of OCS from the oceans as well as indirect release via outgassing of dimethyl sulfide (DMS) and subsequent oxidisation into OCS is highest at mid-latitudes during spring and summer. Indirect oceanic release via outgassing of carbon disulfide ($CS_2$) occurs at low latitudes and maximises during local summer. Vegetative uptake is largest during plant growth

in summer and is much stronger in the northern hemisphere, where the global land masses are concentrated. According to subsequent assessments of Xu et al. (2002), Sandoval-Soto et al. (2005), Montzka et al. (2007), Suntharalingam et al. (2008) and Berry et al. (2013) vegetative uptake might be up to 3–6 times higher than estimated in Kettle et al. (2002). A comprehensive compilation of these budget estimations is given in Kremser et al. (2016).

In most budget estimations, biomass burning has not been considered as a major source of atmospheric OCS (Chin and Davis,

1993; Nguyen et al., 1995; Watts, 2000; Kettle et al., 2002; Montzka et al., 2007; Suntharalingam et al., 2008; Berry et al., 2013). Nguyen et al. (1995) give an estimate of 10% for the contribution of biomass burning to the global OCS emissions, and the contribution of biomass burning in Berry et al. (2013) is about 13%. However, Notholt et al. (2003) measured extraordinarily enhanced OCS amounts of up to 600 pptv on two ship cruises (October/November 1996, December/January 1999/2000) in the height region 10–18 km above the tropical and subtropical Atlantic, which they attributed to biomass burning. Further, Barkley

et al. (2008) presented ACE-FTS observations of enhanced OCS at 10–15 km between the Equator and 30°S during boreal fall, which they assigned to biomass burning as well.

Another topic of scientific interest is the question, if there is a trend in atmospheric OCS or if the budget is balanced. A positive trend would potentially lead to an increase of the stratospheric aerosol layer and thus cause a negative climate forcing. However, since OCS itself is a strong greenhouse gas, the net climate forcing of an OCS trend would likely be very small





(Brühl et al., 2012). For the period 1978–2002, Rinsland et al. (2002) calculated a slow decrease of (-2.5±0.4)%/decade for the altitude range of 2–10 km above Kitt Peak (31.9°N, 111.6°W). Mahieu et al. (2005) reported a decrease of about 7% of the total OCS column above the Jungfraujoch (46.5°N, 8.0°E) obtained by ground-based FTIR spectroscopy between 1988 and 2001, and an increase during the subsequent three years. According to Lejeune et al. (2011) the time series of OCS column

amounts at the Jungfraujoch in Switzerland exhibits a significant positive trend of (11.5±0.6)%/decade during the years 2002 to 2007, but a negative trend of (-11.0±1.9)%/decade for the period 2008–2012. These trends are mainly driven by tropospheric processes and nearly insignificant in the stratosphere. However, especially the last trend estimate has to be considered carefully because of the shortness of the time period. In a long time series of firn air collected near the South Pole Montzka et al. (2004) found a significant increase from the mid 1800s to the late 1900s, but a considerable decrease of 10–16% since the

late 1980s. However, Montzka et al. (2007) did not find a significant tropospheric trend in measurements between 2000 and 2005 at any of the ground-based stations of the global network of the Earth System Research Laboratory of the National Oceanic and Atmospheric Administration (NOAA/ESRL). In another investigation of ground-based FTIR measurements at three stations between 34.45°S and 77.80°S, Kremser et al. (2015) determined positive trends between (4.3±0.2)%/decade and (7.3±0.3)%/decade and concluded that the OCS budget is not closed.

Long time series of OCS are available from a few locations only. Ground-based Fourier transform infrared (FTIR) measurements at the Jungfraujoch in Switzerland started in the mid-1980s (Mahieu et al., 2005). Balloon-borne observations by the MkIV FTIR spectrometer of the Jet Propulsion Laboratory (Toon, 1991) have been performed since 1989, with 23 launches mainly from Fort Sumner, New Mexico, Esrange, Sweden, and Fairbanks, Alaska. Since the year 2000, NOAA/ESRL has performed in-situ measurements of OCS on a global network of about a dozen stations (Montzka et al., 2007). A compilation of

the latitudinal OCS distribution obtained by different measurement techniques (ground-based in-situ and FTIR, balloon-borne FTIR, aircraft and ship measurements) has been presented by Krysztofiak et al. (2014). First spaceborne OCS measurements were performed with the ATMOS interferometer on the space-shuttle (Farmer et al., 1987). A global spaceborne data set of OCS was derived from MIPAS measurements by Burgess et al. (2004) for a restricted time period. Barkley et al. (2008) presented upper tropospheric and stratospheric global OCS distributions of the time period 2004–2006 from spaceborne measurements of

the Fourier Transform Spectrometer of the Atmospheric Chemistry Experiment (ACE-FTS) on the Canadian SCISAT satellite (Bernath et al., 2005; Boone et al., 2005).

  The MIPAS OCS data set has already been presented in Glatthor et al. (2015a), but focused on the signatures of tropical sources and sinks at the 250 hPa level only. In this paper we will give a more global overview of this data set. We will describe the MIPAS instrument, the OCS retrieval setup and the validation of the MIPAS OCS data. Then we will present time series

of the OCS data set at different height levels and discuss latitudinal cross sections as well as seasonal variations. Horizontal distributions at different upper tropospheric pressure levels will indicate source and sink regions of OCS. The contributions of oceanic release and of uptake by tropical vegetation, which have already been presented in Glatthor et al. (2015a), will only be shortly discussed. Instead we will investigate the impact of biomass burning and meridional transport patterns visible in the tropopause region. Further we will perform a dedicated analysis of the decadal trend.





## 2 MIPAS measurements

### 2.1 Instrument description

On March 1, 2002, the ENVIronmental SATellite (ENVISAT) was launched into a Sun-synchronous polar orbit at about 800 km altitude. Among various other experiments the satellite contained the Michelson Interferometer for Passive Atmospheric

Sounding (MIPAS). MIPAS is a limb-viewing Fourier transform infrared emission spectrometer covering the mid-infrared spectral region between 685 and 2410 $cm^{-1}$ (4.1–14.6 $\mu$m), which enables simultaneous observation of numerous trace gases (European Space Agency (ESA), 2000; Fischer et al., 2008). MIPAS data are available from June 2002 until April 2012, when the communication to ENVISAT was lost. MIPAS has been operated in two different measurement modes, the original high resolution (HR) mode from June 2002 to April 2004 and in the reduced resolution (RR) mode since January 2005. This change

of the measurement mode was due to technical problems and resulted in a reduction of the spectral resolution from 0.025 to 0.0625 $cm^{-1}$. On the other hand this change led to reduction of the latitudinal sampling distance from about 530 to 400 km.

We present data of the HR and of the RR "nominal" measurement modes as well as of the RR "UTLS" mode. The nominal modes consisted of rearward limb-scans covering the altitude region between 7 and 72 km within 17 and 27 altitude steps, respectively, and the UTLS mode of scans from about 5.5 to 49 km altitude. The vertical sampling of the HR mode was 3 km

up to 42 km and 5 to 8 km at higher altitudes. The sampling of the RR nominal and UTLS modes was 1.5 km up to 22 km, 2 km up to 32 km, 3 km up to 44 km and 4-4.5 km for the upper part of the scan. MIPAS could be operated during day and night, and nominally produced 1000 and 1400 scans per day in HR and RR nominal mode, respectively. The level-1B radiance spectra used for retrieval are data version 5.02/5.06 (reprocessed data) provided by the European Space Agency (ESA) (Nett et al., 2002).

### 2.2 Retrieval method and error estimation

Retrievals were performed with the common processor of the Institut für Meteorologie und Klimaforschung and the Instituto de Astrofísica de Andalucía (IMK/IAA). This processor uses the Karlsruhe Optimized and Precise Radiative Algorithm (KOPRA) (Stiller, 2000) for radiative transfer calculations and the Retrieval Control Program (RCP) of IMK/IAA for inverse modelling. Processing of MIPAS data at IMK has been described in, e.g., von Clarmann et al. (2003) and Höpfner et al. (2004). The pre-

sented OCS distributions are the most recent data versions, i.e. V5H_OCS_20 of the MIPAS HR mode and V5R_OCS_221/222 as well as V5R_OCS_120 of the MIPAS RR mode.

The OCS retrieval at IMK is performed using 7 microwindows, which cover the spectral range 839–876 $cm^{-1}$ of the fundamental $\nu_1$ band. To model the spectroscopic signature of OCS, line data of the 2009-update for OCS of the HITRAN-2008 database (Rothman et al., 2009) were used. This update is consistent with the OCS spectroscopy in the HITRAN-2012 re-

lease (Rothman et al., 2013). The spectral line shape of the interfering gases $H_2O$, $CO_2$, $O_3$ and $NO_2$ was modelled using the spectroscopy of the MIPAS database, version 3.2 (Flaud et al., 2003). For simulation of the remaining interfering gases the spectroscopy of the HITRAN database was used. The inversion consists of derivation of vertical profiles of atmospheric state parameters from MIPAS level-1B spectra by constrained non-linear least squares fitting in a global-fit approach (von





Clarmann et al., 2003). Since the retrieval grid chosen has a finer altitude spacing than the height distance between the tangent altitudes, a constraint is necessary to attenuate instabilities. For this purpose, Tikhonov's first derivative operator was applied (Steck, 2002, and references therein). Instead of climatological OCS profiles, a height-constant profile was chosen as a priori. As a result, the fitted profile is a smoothed representation of the true profile, but the abundances are not pushed towards

the a priori values. Along with OCS, the profiles of the main interfering trace gases $HNO_3$, CFC-11 and $O_3$ were jointly fitted. Additional retrieval variables were microwindow-dependent continuum radiation profiles and microwindow-dependent, but height-independent zero-level calibration corrections. The radiative contribution of other interfering gases like $H_2O$, ClO, $C_2H_6$, $NO_2$ was modelled by using their profiles retrieved earlier in the processing sequence. The contribution of interfering gases, which had not been prefitted, was modelled using the profiles of the climatology by Remedios et al. (2007).

MIPAS data provide information on atmospheric OCS from the lower end of the profiles in the free troposphere up to about 40 km altitude. To illustrate the performance of a single scan retrieval, Figure 1 (left) shows an OCS profile obtained on 12 July 2009 at southern mid-latitudes along with the measurement noise error. The profile exhibits OCS volume mixing ratios (VMRs) of 425–450 pptv in the troposphere and a strong decline in the stratosphere to values below 100 pptv above 30 km altitude. The vertical resolution (Figure 1, right) degrades from 4–5 km in the troposphere to more than 10 km in the upper

stratosphere. The total estimated error is between 41 and 48 pptv (10–26%) in the troposphere and lower stratosphere and increases to 120 pptv (195%) at 40 km altitude (Table 1). The dominating error component is measurement noise, while the error contributions from interfering species and instrumental parameters are comparatively weak.

## 3   Validation

A dedicated validation of MIPAS OCS amounts measured in the tropical upper troposphere and extra tropical lowermost

stratosphere has been presented in the Supplemental of Glatthor et al. (2015a). In this investigation a comparison has been performed with airborne in-situ data obtained on several campaigns of continental extent or nearly ranging from pole to pole. The outcome was good agreement of MIPAS OCS with the airborne data, mostly with deviations of less than 25 pptv. Here we present a validation for a larger altitude range extending from the troposphere to the middle stratosphere.

### 3.1   Comparison with MkIV balloon interferometer profiles

Figure 2 shows a comparison of MIPAS OCS data with OCS profiles obtained by the balloon-borne MkIV FTIR spectrometer (Toon, 1991). The MkIV profiles were obtained on 16 December 2002 and 1 April 2003 over Esrange, Sweden, (67.9°N, 21.1°E) and on 19 September 2003, 23 and 24 September 2011 over Fort Sumner, New Mexico, U.S., (34.5°N, 104.2°W). Unlike MIPAS retrievals at IMK, evaluation of OCS from MkIV spectra is performed in the spectral region 2041–2077 $cm^{-1}$. The displayed MIPAS profiles are averages of all measurements within a radius of 1000 km and a temporal offset of 24 hours

(48 hours for 19 September 2003) around the tangent points and measurement times of the balloon profiles, which resulted in averaging of 10–14 profiles for the different days. The dotted lines denote the standard deviation of the MIPAS profiles taken into account. On all days except of 16 December 2002 the deviations between the MIPAS and MkIV profiles are less or





equal ±60 pptv, and the OCS values of MIPAS are within the uncertainties of the balloon profiles at nearly all altitudes. On 16 December 2002 there are larger differences of up to 90 pptv at 22 km altitude, where the balloon profile exhibits a sharp decrease to 0 pptv. However, the MIPAS profile shows a similar decrease only slightly above at 23 km altitude. In summary, no clear indication of a bias against MkIV profiles has been found.

## 3.2 Comparison with SPIRALE balloon in-situ profiles

In Figure 3 we present an intercomparison of MIPAS OCS measurements with balloon-borne in-situ measurements by the SPectromètre InfraRouge d'Absorption à Lasers Embarqués (SPIRALE), which were performed on 9 June 2008 over Teresina, Brazil (5.1°S, 42.1°W) and on 24/25 August 2009 over Esrange, Sweden (67.9°N, 21.1°E) (Krysztofiak et al., 2014). The instrumental concept consists of six mid-infrared laser beams, which are reflected between two mirrors in a multipass cell located below the gondola (Moreau et al., 2005). OCS was retrieved in the wavenumber region 2056.1–2056.5 $cm^{-1}$ using the HITRAN12 spectroscopy (Rothman et al., 2013). Again, the displayed MIPAS profiles are averages of all measurements within a radius of 1000 km and a temporal offset of 24 hours around the tangent points and measurement times of the balloon profiles, which resulted in averaging of 18 and 26 profiles for the Teresina and Kiruna flight, respectively. Since the original SPIRALE profiles are vertically much better resolved than the MIPAS profiles, they are much more structured. For comparison of the two experiments the balloon profiles have additionally been convolved with the MIPAS averaging kernels. On both days there is good agreement between the two instruments. On 9 June 2008 the deviations are between -50 pptv at 16 km and 15 pptv at 26 km altitude. On 24/25 August 2009 the differences range between 30 pptv at 15 and -10 pptv at 18 km altitude. Thus, there is also no clear indication of a bias against SPIRALE profiles.

## 3.3 Comparison with ACE-FTS

Another global OCS data set was derived from spaceborne measurements of the ACE-FTS experiment on SCISAT (Bernath et al., 2005; Boone et al., 2005). Different to the MIPAS retrieval setup, the region 2039–2057.5 $cm^{-1}$ is used for ACE-FTS retrievals of OCS (Hughes et al., 2016), and the linelist of the HITRAN 2004 database (Rothman et al., 2005) is applied. In Figure 4 we show an intercomparison of mean OCS profiles of coincident MIPAS and ACE-FTS observations. The ACE-FTS profiles taken into account are data version v3.5 (Boone et al., 2013). The comparison is based on 9452 ACE-FTS and on 19990 MIPAS profiles of the time period February 2004 to April 2012. For each ACE-FTS profile all MIPAS profiles within a maximum temporal offset of 5 hours and a maximum spatial distance of 500 km were taken into account. Then all coincident profiles of the two data sets were averaged for six different 30-degree wide latitude bands. The profiles obtained by the two instruments exhibit significantly different shapes. While the MIPAS OCS volume mixing ratios slightly increase from the middle to the upper troposphere before they strongly decrease in the stratosphere, the ACE-FTS profiles in most latitude bands continuously decrease with altitude. Consequently, there is a positive bias between MIPAS and ACE-FTS in the height region 8–20 km. The differences are largest in the altitude region 13–14 km, where they amount to 75–100 pptv. There is rather good agreement at the lower ends of the profiles and above 20 km altitude.




The reason for the large differences between the OCS amounts obtained by MIPAS and ACE-FTS is unclear. A possible explanation is the use of different spectral regions for OCS retrieval (see above). However, in this case it is difficult to explain the vanishing deviations below 10 and above 20 km altitude. In support of the higher MIPAS OCS values we refer to Velazco et al. (2011), who found ACE-FTS OCS data to be biased low by ∼15% as compared to MkIV results in the altitude range 14–23 km, although both data sets are retrieved in the same spectral region. Further, Krysztofiak et al. (2014) reported on a low bias of ACE-FTS OCS as compared to the SPIRALE profiles presented in Section 3.2, which are also retrieved in the same spectral region as used for ACE-FTS analysis of OCS. The mean ACE-FTS profile with respect to the SPIRALE profile of Teresina is 15–20% lower below 22.5 km altitude and the mean ACE-FTS profile in relation to the Kiruna flight is 20% lower at 16.5 km altitude. In addition, the validation results of Velazco et al. (2011) and Krysztofiak et al. (2014) are also in agreement with our intercomparison of MIPAS OCS data with MkIV and SPIRALE measurements.

## 4 Results and discussion

### 4.1 Data set overview

As a general overview, Figure 5 shows time series of monthly zonal averages of OCS at 10–30 km altitude, covering the operational period of MIPAS from July 2002 to April 2012. Spatial averaging was performed for $7.5° \times 1$ km latitude-altitude bins at the poles and $5° \times 1$ km bins elsewhere. This resulted in summation of generally 500–1000 values per month in each latitude-altitude bin except for the altitudes of 10 and 14 km in the inner tropics, where the number of points per bin decreases significantly due to strong cloud contamination, which reduces the number of usable measurements. Thus the tropical bins at 10 km altitude contain 69 individual data points on average and no measurements at all at the beginning of 2011. Caused by contamination by polar stratospheric clouds the number of data points can also be considerably reduced inside the polar vortices. The data gap during most of the year 2004 results from the operational shutdown between MIPAS HR and RR measurement periods.

At most altitudes there is no distinct bias between the OCS amounts of the HR and of the RR period, indicating consistent OCS amounts over the whole operational period. However, at 14 km the OCS amounts of the RR period are significantly higher (by ∼25 pptv in the tropics, by ∼50 pptv at mid- and high latitudes) than those of the HR period. A closer look shows that the period of higher OCS amounts does not coincide with the beginning of the RR mode immediately after the data gap, but starts some months later in June 2005. At 18 km there is also an increase in tropical OCS, but not before the year 2006. Thus this increase might have a geophysical reason. Apart from these offsets there is no indication of a strong trend in the time series. A dedicated trend analysis will be performed in Section 4.4.

Generally, there are no strong interannual variations at all altitudes. At 10 km there is a clear annual variation in the northern hemisphere with maxima around May/June in the tropics and around September/October at high latitudes. In the tropics the lowest OCS values were measured at the end of the year and at Arctic latitudes around March/April. In the southern hemisphere there is a less distinct annual cycle, and the maxima are somewhat weaker. The OCS amounts peak around July at low southern latitudes and around May at southern mid-latitudes. The OCS amounts at the altitude of 14 km also exhibit annual cycles at





northern as well as at southern latitudes. In the southern hemisphere the annual cycle at this altitude is more distinct than at 10 km. While the magnitude of low- and mid-latitude OCS amounts at 14 km is approximately the same as at 10 km until mid-2005, it is higher in the subsequent time period. The Arctic minima between December and March and the Antarctic minima from March until the end of the year are obviously caused by subsidence of OCS-poor air masses in the polar vortices.

At 18 km the annual variations at low and mid-latitudes become weaker, while high latitudes are also characterised by vortex dynamics. At 22–30 km the OCS variations are further weakened and in the tropics the 1-year cycle is superposed by a 2-year period, indicating the impact of the quasi-biennial oscillation (QBO).

Figure 6 shows MIPAS OCS profiles of the latitude bands 0°–30°, 30°–60° and 60°–90° both for the northern and southern hemisphere. The profiles are seasonal averages of the whole OCS data set. Up to 13–15 km the mixing ratios vary only little
between about 500 and 530 pptv. In the stratosphere they strongly decrease to below 100 pptv around 20 km at high latitudes and at 30 km in the tropics. The largest seasonal variations occur in the Arctic and Antarctic lower stratosphere, reflecting subsidence of OCS poor air masses in the polar vortices during winter and spring and the recovery during summer and fall. The subsidence is stronger and continues until Austral spring in the Antarctic vortex.

The seasonal variations in the troposphere are generally larger in the northern hemisphere. In the northern tropics and
subtropics the OCS amounts between 7 and 12 km are 510–530 pptv in boreal spring and summer and about 20–30 pptv lower during boreal fall and winter. The highest OCS amounts of 500–520 pptv at northern mid-latitudes were measured during boreal summer. At northern high-latitudes the tropospheric maximum is temporally further shifted into boreal fall. These variations are caused by overlap of oceanic release and of vegetative uptake of OCS during the growing season, which affects the free troposphere with a delay of some months. At southern low and mid-latitudes there are only small tropospheric variations, and
the mixing ratios in austral summer are somewhat lower than those in the northern hemisphere during boreal summer.

## 4.2   Latitudinal cross sections

### 4.2.1   Annual means

Since OCS is assumed to be relatively well mixed in the troposphere, we compared the latitudinal variation of MIPAS OCS amounts at 8 km altitude with ground-based flask data from the NOAA/ESRL network (Montzka et al., 2007). The result
is shown in Figure 7, where both data sets are averaged over the whole measurement period of MIPAS. This kind of temporal averaging eliminates a potential bias due to the time required for vertical transport of seasonal variations to the upper troposphere. Between southern mid-latitudes and northern subtropics there are differences of less than 10 pptv. The largest deviations occur at northern mid-latitudes, where the OCS amounts at three ground-based stations in the United States are lower by 25–50 pptv. According to Montzka et al. (2007), these low amounts are caused by vegetative uptake. Obviously this
signal is hardly observable in zonal means in the upper troposphere. However, there is much better agreement of MIPAS OCS with the high-altitude station Niwot Ridge (40.0°N, 105.54°W, 3475 m asl), Colorado, where a decrease caused by vegetative uptake is also not visible. There is also better correspondence of MIPAS data from northern mid-latitudes with surface OCS data from Mace Head (53.3°N, 9.9°W, 42 m asl) in Ireland, where a potential signal of vegetative uptake advected from North





America by the prevailing westerly winds is mixed with other air masses on its long way crossing the Atlantic Ocean. Due to the stronger impact of vegetative uptake on measurements in the boundary layer, the OCS amounts from two stations near sea-level at high northern latitudes (Barrow and Alert) are also considerably lower than those observed by MIPAS at 10 km. Again, there is better agreement with OCS amounts measured in the free troposphere at the station Summit (72.6°N, 38.4°W,

∼3200 m asl) on top of the Greenland ice shield. At high southern latitudes the ground-based OCS amounts remain on the same level as at mid-latitudes, while the OCS values at 10 km altitude exhibit a step-like decrease.

### 4.2.2 Seasonal variations

Figure 8 (left column) shows latitude-height cross sections of MIPAS OCS VMRs measured in winter, spring, summer and fall. The seasonal data sets are averaged over the whole measurement period of MIPAS (2002–2012). Averaging was performed

for $7.5° \times 1$ km latitude-altitude bins at the poles and $5° \times 1$ km bins elsewhere. Above 10 km altitude the averaged MIPAS data are generally based on 10000–15000 values. Due to cloud-contamination and to the upward-shift of the MIPAS RR mode scans towards low latitudes, increasingly less data points could be binned at 10 km and below, e.g., only some dozens or even less than 10 values at 7 km altitude in the tropics.

Because of the long lifetime of OCS, the latitude-height cross sections of measured OCS appear rather similar through all

seasons. Tropospheric OCS amounts are between 480 pptv at high latitudes and 520 pptv in the tropical upper troposphere. Further, the tropospheric cross sections are relatively symmetric with respect to the Equator. At mid- and high latitudes there is a strong vertical decline in OCS in the lower stratosphere. At tropical latitudes this transition region is shifted towards the middle stratosphere and vertically extended, which reflects upwelling in the Brewer-Dobson circulation. During winter and spring subsidence of stratospheric air masses with low OCS amounts is visible in the polar vortices. This process is more

pronounced in the Antarctic vortex as compared to the Arctic vortex.

For comparison, Figure 8 (second column) shows cross sections of an ECHAM/MESSy Atmospheric Chemistry (EMAC) model run. Horizontal distributions of this simulation have already been presented in Glatthor et al. (2015a). The EMAC model is a numerical chemistry and climate simulation system that includes sub-models describing tropospheric and middle atmosphere processes and their interaction with oceans, land and human influences (Jöckel et al., 2006, 2010). It uses the second

version of the Modular Earth Submodel System (MESSy2) to link multi-institutional computer codes. The core atmospheric model is the 5th generation European Centre Hamburg general circulation model (ECHAM5, Roeckner et al., 2006). More details of the model simulation can be found in Glatthor et al. (2015a).

Generally, the model cross sections are in good agreement with the measurements. They also show upwelling in the tropics, decreasing values in the stratosphere and towards higher latitudes and subsidence in the polar vortices, but there are several

differences. First, contrary to the measurements the modelled tropospheric OCS amounts are not symmetrical in relation to the Equator. They are in rather good agreement with the measurements in the southern hemisphere, but decrease towards high latitudes in the northern hemisphere. This asymmetry is most pronounced in boreal fall and due to the strong vegetation uptake assumed in the model run. Secondly, the modelled transition region from high tropospheric to low stratospheric mixing ratios is at somewhat higher altitudes than in the measurements. Thirdly, the MIPAS distributions generally exhibit a stronger upwelling



in the tropical stratosphere above 30 km than the model distributions. To investigate if these differences are at least partly due to the observation conditions, the EMAC profiles were convolved with the averaging kernel of a MIPAS OCS profile obtained in the tropics (right column). This procedure leads to much better agreement in the tropical upper stratosphere and to somewhat better agreement in the northern hemispheric troposphere and in the transition region between 15 and 25 km altitude.

### 4.2.3 Mean hemispheric mixing ratios

In Table 2 we show seasonal and annual averages of MIPAS OCS measurements in the northern and southern hemispheric troposphere. For this purpose the OCS VMRs in all zonal bins below the mean tropopause were averaged separately for both hemispheres. Tropopause heights were calculated from the temperature profiles obtained from MIPAS measurements and averaged in the same way as the OCS measurements. In both hemispheres the seasonal variation of average upper tropospheric OCS is only 10–11 pptv. The interhemispheric differences are also very low, leading to ratios close to unity. The ratio between the annual means of the northern and southern hemisphere is $1.01\pm0.01$. The same ratio results for all seasons except of boreal spring, where it is 0.99. For comparison, the ratio between the OCS amounts observed by ACE-FTS during 2004–2006 in the northern and southern hemispheric troposphere is 1.03 (Barkley et al., 2008). The annual NH/SH ratio obtained from ground-based measurements of the NOAA/ESRL network between 2000 and 2005 is slightly lower than unity, namely $0.97\pm0.01$ (Montza et al., 2007). In the course of the year this ratio varies between 1.02 in boreal spring, 0.98 in boreal summer, 0.92 in boreal fall and 0.95 in boreal winter. In this estimation all northern hemispheric sites situated at more than 3000 m above sea level had been excluded. Additional omission of the two northern hemispheric mid-continental sites led to an annual mean NH/SH ratio of 1.00, which is in good agreement with the ratio for MIPAS OCS and indicates that the interhemispheric imbalance is caused by the strong vegetation uptake of OCS observed at these two sites during summer and autumn (Montzka et al., 2007).

### 4.2.4 Monthly variations in different latitude bands

For a more quantitative investigation of seasonal variations, Figure 9 shows monthly averages of measured OCS amounts in six 30°-wide latitude bands at 10 km altitude along with the monthly averages of all ground stations of the NOAA/ESRL network in the respective latitude band. In both data sets there are distinct seasonal variations in all latitude bands, which generally are larger at the ground than at 10 km altitude. Further, the variations in the northern hemisphere are stronger than those at southern latitudes.

Due to the strong seasonal cycle in northern hemispheric vegetation uptake, the largest variations of up to more than 100 pptv occur at northern hemispheric mid- and high latitude ground-based stations. The OCS amounts observed at these stations peak in April/May just before the onset of vegetative uptake and are lowest in September. The respective variations in MIPAS OCS at 10 km altitude are 30–40 pptv and exhibit a phase lag of 4–6 months, indicating a relatively slow upward propagation of these mid-latitude surface variations. A similar behaviour, namely reduced amplitudes and a time delay of 1–2 months of OCS variations in the free troposphere as compared to surface stations, was also reported by Montzka et al. (2007) for airborne measurements above the United States. In the northern tropics and subtropics there is a temporal shift of one month at the most





between measured ground-based and upper tropospheric variations, which apparently reflects faster upward propagation than at higher latitudes.

Ground-based OCS amounts in the southern hemisphere have peak-to-peak amplitudes of about 35 pptv, with maxima in February/March and minima in August/September. The respective variations observed by MIPAS at 10 km altitude are 15–20

5   pptv. Compared to the surface measurements the southern hemispheric maxima observed by MIPAS are delayed by three to four months.

### 4.3 Horizontal distributions

#### 4.3.1 Tropical sources and sinks, meridional decline

Figure 10 shows OCS amounts measured at 250 hPa during June to August (top left) and September to November (top right).

This pressure level corresponds to $\sim$11 km in the tropics and $\sim$9.7 km at polar latitudes. The distributions are averaged over the whole measurement period of MIPAS from 2002 until 2012. The latitude-longitude binning is $7.5° \times 15°$ at the poles and $5° \times 15°$ at lower latitudes. The complete seasonal variation at this pressure level has already been presented in Glatthor et al. (2015a).

During June to August elevated OCS amounts of up to 540 pptv were observed over wide tropical and subtropical areas

and even up to northern mid-latitudes, apparently caused by strong oceanic release and effective upward transport. During September to November the OCS amounts measured at low latitudes are reduced by 20–30 pptv. Another striking feature is a large area of low OCS amounts extending from Brazil over the southern tropical Atlantic to southern Africa, which is caused by uptake by the tropical vegetation and strongest in austral summer (cf. Glatthor et al., 2015a).

During boreal summer the OCS amounts measured at 250 hPa exhibit a distinct decrease towards mid- and high latitudes,

which is especially pronounced in the southern hemisphere. This decrease is partly due to the fact that at high latitudes this pressure level is in the tropopause region or even in the lowermost stratosphere. But it also indicates that the main source regions of OCS are at low latitudes or that upward transport by strong convection is more effective at these latitudes. During boreal fall the decrease towards high southern latitudes is nearly the same, while the OCS amounts at mid and high northern latitudes have increased, indicating northward transport of elevated OCS amounts observed at lower latitudes during summer

or slower convection.

#### 4.3.2 Biomass burning

As outlined in Section 1 the role of biomass burning as source of atmospheric OCS is controversial, although most budget estimations assume a minor contribution of biomass burning only. Because of this open question we checked the MIPAS OCS distributions on signatures of biomass burning by comparison with MIPAS HCN, which is an almost unambiguous tracer of

biomass burning (Li et al., 2003; Singh et al., 2003). Typical emission factors (g kg$^{-1}$) of HCN for tropical biomass burning are 0.42 (0.26) for tropical forests (Akagi et al., 2011) and 0.53 (0.15) for savanna fires (Yokelson et al., 2003). The values in parentheses are estimates for the natural variation. OCS emission factors for tropical biomass burning have hardly been



determined. Based on one canister sample Yokelson et al. (2008) give a value of 0.025 g kg$^{-1}$, which is more than one order of magnitude lower than the HCN emission factors and thus also points to comparably low OCS emissions from tropical fires.

A prominent feature of biomass burning is the southern hemispheric plume, which forms in boreal fall and mostly extends from Brazil over southern Africa to Australia and further eastward above the southern Pacific (Edwards et al., 2006; Coheur et al., 2007; von Clarmann et al., 2007; Glatthor et al., 2015b). This plume is reflected in strongly enhanced amounts of HCN observed during September to November 2002–2011 (Figure 10, bottom right), but it is not visible in the MIPAS OCS distribution (Figure 10, top right). On the contrary, the area between Brazil and southern Africa contains depleted OCS amounts instead. During boreal summer, signatures of biomass burning are generally detectable inside the Asian Monsoon Anticyclone, over central and western Africa and the north-western subtropical Pacific. Again, this is reflected in enhanced HCN amounts detected by MIPAS in these regions during June to August 2002–2011 (Figure 10, bottom left). However, the distribution of enhanced OCS amounts observed by MIPAS during boreal summer appears rather different (Figure 10, top left). It is much more widespread over the tropics and subtropics and apparently has strong sources around Indonesia and in the western Pacific.

For a more quantitative estimation we performed correlation analyses between upper tropospheric MIPAS OCS and MIPAS HCN for different years, seasons and latitude bands. For illustration, Figure 11 shows correlations of data observed in the northern tropics and subtropics between January and August 2007 (left) and in the southern tropics to mid-latitudes between September and December 2006 (right). The displayed data points are from the altitude range 10–14 km. The first period chosen is characterised by strong northern and the second one by intense southern hemispheric biomass burning (cf. Glatthor et al., 2015b). In both cases the correlation coefficient is around zero. Background HCN amounts scatter between 100 and 400 pptv, while the HCN VMRs in air masses polluted by biomass burning increase up to 800 and to more than 2000 pptv, respectively. This increase is not at all accompanied by a rise in OCS in the northern hemisphere (very weak negative slope instead) and by a weak rise in the southern hemisphere (slope of 0.0227).

We checked the whole OCS data set measured by MIPAS for potential signatures of biomass burning on a monthly basis, but did neither find strong correlations with, e.g., HCN or CO nor especially enhanced OCS in the regions of typical biomass burning plumes. Table 3 shows the results of the correlation analysis for southern hemispheric biomass burning periods (50°S–0°N, September–December) during all years with available MIPAS data. It is evident, that the slight positive slope observed in 2006, which would indicate a small increase of OCS due to biomass burning, is not typical for most of the years of MIPAS observations. Except of the years 2002 and 2003, for which also positive slopes of 0.062 and 0.014 were determined, all other years exhibit negative slopes. Thus we conclude that compared to other sources biomass burning plays only a minor role in the tropospheric OCS budget. The findings of Notholt et al. (2003) and Barkley et al. (2008) (see Section 1) can not be confirmed on the basis of MIPAS data.

A possible reason for the positive slopes in 2002 and 2006 is intense biomass burning in Indonesia during these years, which were characterised by a strong positive phase of the El Niño-Southern Oscillation (ENSO) (http://www.cpc.ncep.noaa.gov/products/analysis_monitoring/ensostuff/ensoyears.shtml). Biomass burning in these regions is characterised by a high percentage of peat fires, for which Akagi et al. ( 2011) give a much higher OCS emission factor of 1.2 (2.21) g kg$^{-1}$.



### 4.3.3   The Asian Monsoon Anticyclone

Figure 12 (top left) shows the OCS distribution observed by MIPAS during June to August at the 150 hPa (∼13.0–14.3 km) level, where enhanced OCS amounts also prevail over wide tropical and subtropical areas between about 30°S and 40°N. Different to the 250 hPa level, the OCS values in the southern tropical and subtropical latitude band are generally of the same amount as those observed at northern low latitudes. The highest OCS amounts were observed above the Middle-East, encircled by a region of enhanced OCS extending from the northern subtropical Atlantic over northern Africa and South-Asia to the Chinese coast. This feature reflects the Asian Monsoon Anticyclone (AMA), which regularly forms in boreal summer in the upper troposphere and usually is situated between northern Africa and eastern China. Enhanced OCS in this region is consistent with observation of high amounts of other trace gases like CO (Funke et al., 2009) or HCN (Randel et al., 2010; Glatthor et al., 2015b), which are effectively transported upward in the region of the Asian summer monsoon and then accumulate in the AMA due to confinement by the strong anticyclonic circulation. A possible reason for the high amounts of OCS inside the AMA is strong release from the northern tropical Pacific and Indian Ocean. Additional potential sources are anthropogenic emissions of OCS, which according to Campbell et al. (2015) are currently concentrated in Asia. While CO or HCN in the AMA region are strongly enhanced, the OCS VMRs are not much higher than above the northern Pacific and above the whole southern tropics. This is due to different sources and the longer lifetime of OCS as compared to the other species.

Instead of the coherent OCS depletion between Brazil and southern Africa observed at 250 hPa there are three areas of reduced OCS at 150 hPa, situated above central America/northern South America, central Africa and north of Indonesia. The southern parts of the American and Indonesian depletions are probably caused by vegetative uptake, while the northern parts seem to be due to meridional transport around the upper tropospheric anticyclones discussed in Section 4.3.4. This assumption is based on comparison with the respective ozone distribution measured by MIPAS (bottom left), which exhibits low tropospheric values in the tropics and considerably higher stratospheric values at mid- and high latitudes. There are two tongues of enhanced $O_3$ extending southwestward towards Central America and towards the Philippines, which also reflect meridional transport and coincide with the northern parts of the OCS depletions. Due to subsidence in the Antarctic vortex, very low OCS amounts of 390–450 pptv were observed at southern high latitudes, while the values in the Arctic are around 490 pptv.

### 4.3.4   Meridional transport in the UTLS region

The OCS distribution observed during boreal summer at the 80 hPa level (∼17.0–17.9 km) (Figure 12, top right) also exhibits the highest values in the tropics and subtropics. However, at this pressure level the major part of the area of enhanced OCS is situated in the southern hemisphere. In the northern hemisphere there is a band of enhanced values extending from the tropical Atlantic to the Chinese coast, which reflects the upper end of the Asian Monsoon Anticyclone including westward outflow. Another region of high OCS values is situated above the eastern Pacific. Directly south of the enhanced OCS amounts at the top of the AMA there is a large region of low values above South and South-East Asia, extending westward as far as West Africa in a small band. These low OCS amounts apparently originate from northern mid-latitudes and were transported




southward and subsequently westward by the anticyclonic circulation at the edge of the AMA. Meridional transport around the AMA is corroborated by, e.g., the ozone distribution measured by MIPAS (bottom row), which in the UTLS region exhibits a tongue of enhanced values reaching from the western Pacific to South-Asia and northern Africa. A similar structure has also been found in ozone data of the Atmospheric Infrared Sounder (AIRS) averaged over July and August 2003 (Randel and Park, 2006, Fig.1d). Further, this transport process was modelled by, e.g., Ploeger et al. (2012) by back trajectory calculations. The major part of modelled southward transport into the tropical latitude band occurred during August to September at potential temperature levels of 420–450 K (∼16.5–18 km), which is in good agreement with our observations.

A weaker, but similar feature of meridional OCS transport is also visible over Central America. The air masses with low OCS amounts detected in this area were obviously transported southward at the eastern side of the quasi-stationary anticyclone centered above the south-western United States. This feature has also been observed in airborne in-situ $O_3$ measurements performed between the south-eastern United States and Central America in July 2002 (Richard et al., 2003). Towards high latitudes the measured OCS amounts decrease strongly.

### 4.4 Trends

A dedicated trend analysis of MIPAS OCS was performed for $10° \times 1$km latitude-altitude bins on the basis of monthly zonal averages. For each bin the analysis was carried out by fitting of regression functions consisting of a linear trend, 8 sine and cosine waves with periods between 3 and 24 months and of two terms describing the quasi-biennial-oscillation to the time series of monthly averages (von Clarmann et al., 2010; Stiller et al., 2012). Figure 13 shows the measured OCS time series and the fitted regression curves exemplarily for two bins at 10 km altitude. The whole set of fitted linear trends is shown in Figure 14 (left) along with the respective significance estimates (right), which were calculated by division of the regression coefficients by their standard deviations. Values of 2 or larger characterise highly significant trends. Except of rather weak trends of about +10 pptv/decade in the tropics and at 50–60°N (∼2%/decade) as well as of -10 pptv/decade (∼-2%/decade) in the southern subtropics and mid-latitudes, no significant temporal change was determined for the altitude of 10 km. According to this result, tropospheric sources and sinks of OCS should be largely balanced. Thus, the significant positive trends above the Jungfraujoch and in the southern hemisphere (cf. Introduction) obtained from ground-based FTIR measurements (Lejeune et al., 2011; Kremser et al., 2015) can not be confirmed.

Considerably stronger trends were calculated for the middle stratosphere. There is a large area of negative trends of up to more than -60 pptv/decade in the whole northern hemisphere, extending from 15 to 30 km in the Arctic and from 20 to 40 km at 10–20°N. On the basis of a 2-$\sigma$ confidence threshold, these trends are highly significant. In the southern hemisphere there is an area of somewhat weaker positive trends extending from 15 to 40 km at mid-latitudes and from about 20 to 30 km in the tropics and in the Antarctic. These trends are also highly significant. A possible reason for theses trends is a change in the global circulation, e.g. in the strength or latitudinal centering of the Brewer-Dobson circulation, during the years of MIPAS operation. Similar structures were detected in trend analyses of, e.g., MIPAS CFC-11 (Kellmann et al., 2012) or ozone (Eckert et al., 2014). Additional features in the trend analysis are regions of negative trends in the tropical upper stratosphere above 40 km and at high southern latitudes between 14 and 20 km altitude. However, these trends are less significant.




## 5 Conclusions

We have presented a long-term spaceborne global data set of MIPAS OCS covering the period from June 2002 to April 2012, providing information on one of the major sources of atmospheric sulfur. This data set can be used to constrain future model runs and thus can help to improve the understanding of the global sulfur cycle. In fact, a paper on this topic, using also MIPAS OCS and $SO_2$ data, is in preparation (Deshler, personal communication). The measurements were compared with balloon profiles, ACE-FTS data and with ground-based measurements of the NOAA/ESRL network. No systematic bias was detected with respect to balloon profiles, but MIPAS OCS amounts at 13–16 km altitude are up to 100 pptv higher than coincident ACE-FTS values. The reason of this discrepancy is an open issue.

Time series of MIPAS OCS data show annual variations in the troposphere and a rather biennial cycle in the tropical strato-sphere. Interannual variations are only moderate, indicating a closed budget of tropospheric OCS sources and sinks. This is confirmed by a dedicated trend analysis, which resulted in no significant trends in the upper troposphere. In the stratosphere, a large area of strong negative trends was detected in the northern hemisphere accompanied by an area of positive trends in the southern hemisphere. This pattern indicates a change in the Brewer-Dobson circulation.

Zonal averages of tropospheric OCS amounts observed by MIPAS range from below 470 pptv at high latitudes to 540 pptv in the tropics and at northern mid-latitudes, but show only moderate seasonal variations. These features are in reasonably good agreement with EMAC model simulations. Interhemispheric differences are small, e.g. the ratio between northern and southern hemispheric annual tropospheric averages is 1.01 only. At 10 km altitude, zonally averaged seasonal variations have an amplitude of 30–35 pptv in the northern and of 15–20 pptv in the southern hemisphere. The variations at northern mid- and high latitude surface stations of the NOAA/ESRL network are much higher, namely up to 110 pptv. This is due to the strong seasonal variation of vegetation uptake influencing these measurements. Obviously this signal is considerably weakened in zonal means at 10 km altitude. In the southern hemisphere the amplitudes observed at the surface stations and at 10 km altitude are lower and in better agreement.

At northern hemispheric low latitudes maximum MIPAS OCS amounts were measured around June and minima in fall and winter, which fairly well agrees with the seasonality of surface observations. At northern mid- and high latitudes maxima were observed in August and October and minima in March and April. These variations are temporally delayed by 4–6 months compared to the ground-based cycles, where the OCS decrease starts in May and June. A possible reason for the larger temporal shift at higher latitudes is weaker convection than at tropical latitudes, but meridional transport processes can also play a role. In the southern hemisphere there are temporal shifts of 3–4 months between the variations observed at the ground and at 10 km altitude at all latitudes.

The stratospheric OCS distribution is similar to that of other trace gases like $N_2O$ or CFC-11. Due to broad upwelling the highest stratospheric OCS values were observed in the lower tropical stratosphere. Towards higher altitudes and latitudes stratospheric OCS exhibits a considerable decrease. Low stratospheric OCS amounts at high latitudes during polar winter indicate subsidence of air masses inside the polar vortices.





Horizontal distributions of upper tropospheric OCS exhibit enhanced OCS amounts at low latitudes, most pronounced during boreal summer above the western Pacific and the Indian Ocean. Apparently, these enhancements are mainly caused by oceanic release. A striking feature is a region of low OCS amounts between Brazil and southern Africa, which is most distinct during austral summer. This depletion obviously is caused by vegetative uptake in the tropical forests. Typical features of biomass
burning like the southern hemispheric plume are not visible in MIPAS data, indicating that biomass burning is only a minor source of atmospheric OCS. The enhanced amounts of OCS observed in the Asian Monsoon Anticyclone rather result from oceanic and industrial emissions. The OCS distributions at 80 hPa altitude reflect equatorward transport of mid-latitude air masses around summer anticyclones prevailing in the upper troposphere. This process is most distinct at the western and southern edge of the Asian Monsoon Anticyclone, but also observable in the western hemisphere, where air masses deleted in
OCS are carried towards Central America.

*Acknowledgements.* The authors like to thank the European Space Agency for giving access to MIPAS level-1 data. Meteorological analysis data have been provided by ECMWF. We acknowledge support by the Deutsche Forschungsgemeinschaft and Open Access Publishing Fund of the Karlsruhe Institute of Technology. The Atmospheric Chemistry Experiment (ACE), also known as SCISAT, is a Canadian-led mission mainly supported by the Canadian Space Agency and the Natural Sciences and Engineering Research Council of Canada. We acknowledge
the provision of MkIV and SPIRALE balloon profiles.



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



**Table 1.** Error budget for an OCS retrieval from spectra recorded on 12 July 2009 at 42.8°S, 18.7°E for selected altitudes. The errors are given in units of pptv, values in parentheses are relative errors in %. The total error is the sum of measurement noise and parameter error. The parameter error is the sum of uncertainties in interfering species and instrumental properties.

| Height km | Total Error | Measurement Noise | Parameter Error | $NH_3$ | Line of Sight | Gain | Shift | Temperature |
|---|---|---|---|---|---|---|---|---|
| 10 | 45 (10.0) | 40 (8.9) | 21 (4.7) | 17 (3.8) | 6.5 (1.4) | 6.6 (1.5) | 1.1 (0.2) | 0.54 (0.1) |
| 15 | 41 (9.9) | 38 (9.2) | 14 (3.4) | 3.4 (0.8) | 12 (2.9) | 4.5 (1.1) | 0.76 (0.2) | 0.60 (0.1) |
| 20 | 48 (26.3) | 46 (25.2) | 12 (6.6) | 6.1 (3.3) | 8 (4.4) | 2.2 (1.2) | 4.0 (2.2) | 0.36 (0.2) |
| 25 | 60 (46.9) | 59 (46.1) | 6.4 (5.0) | 4.6 (3.6) | 2.3 (1.8) | 0.98 (0.8) | 2.3 (1.8) | 0.09 (0.1) |
| 30 | 73 (125) | 73 (125) | 5.5 (9.4) | 4.8 (8.2) | 0.45 (0.8) | 0.91 (1.6) | 2.1 (3.6) | 0.09 (0.1) |
| 40 | 120 (195) | 120 (195) | 8.4 (13.7) | 7.3 (11.9) | 2.4 (3.9) | 0.84 (1.4) | 3.0 (4.9) | 0.07 (0.1) |

**Table 2.** Seasonal and annual averages of OCS amounts measured by MIPAS between June 2002 and April 2012 in the northern and southern hemispheric troposphere and ratio between hemispheric averages. MAM: March to May, JJA: June to August, SON: September to November, DJF: December to February.

| Season | Average OCS Northern Hemisphere | amounts in pptv Southern Hemisphere | Ratio NH/SH |
|---|---|---|---|
| MAM | 493.51 | 496.46 | 0.99±0.01 |
| JJA | 502.03 | 495.60 | 1.01±0.01 |
| SON | 498.51 | 495.51 | 1.01±0.01 |
| DJF | 491.62 | 484.88 | 1.01±0.01 |
| Annual | 494.42 | 491.37 | 1.01±0.01 |





**Table 3.** Results of a correlation analysis between MIPAS OCS and simultaneously measured HCN for the latitude band between $50°$S– $0°$N and the altitude range 10–14 km during southern hemispheric biomass burning seasons of subsequent years; r: correlation coefficient; slope: slope of the regression line, the values in parentheses are the 1-$\sigma$ uncertainties.

| Period | r | slope |
|---|---|---|
| Sep-Dec 2002 | 0.154 | 0.062 (0.0015) |
| Sep-Dec 2003 | 0.017 | 0.014 (0.0027) |
| Sep-Dec 2005 | 0.000 | -0.001 (0.0030) |
| Sep-Dec 2006 | 0.076 | 0.027 (0.0019) |
| Sep-Dec 2007 | -0.08 | -0.041 (0.0018) |
| Sep-Dec 2008 | -0.09 | -0.075 (0.0024) |
| Sep-Dec 2009 | -0.02 | -0.016 (0.0021) |
| Sep-Dec 2010 | -0.06 | -0.029 (0.0016) |
| Sep-Dec 2011 | -0.02 | -0.015 (0.0018) |





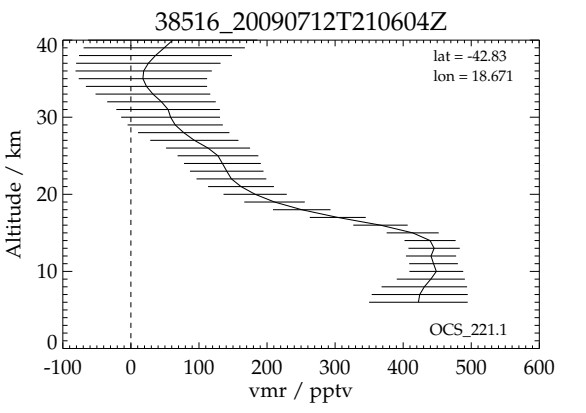
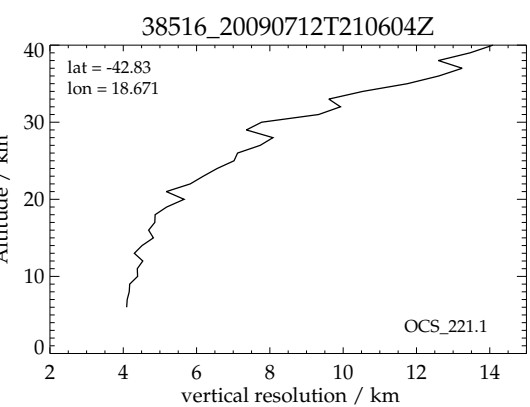

**Figure 1.** Left: MIPAS OCS profile obtained from measurements at southern mid-latitudes (42.8°S, 18.7°E) on 12 July 2009 along with the measurement noise error. Right: respective vertical resolution, derived from the width of the rows of the averaging kernel.







**Figure 2.** Comparison of MIPAS OCS data (black) with OCS profiles obtained by the MkIV balloon experiment (red) on 16 December 2002 and 1 April 2003 over Esrange, Sweden, (67.9°N, 21.1°E, top panel) and on 19 September 2003, 23 and 24 September 2011 over Fort Sumner, New Mexico (34.5°N, 104.2°W, middle panel and bottom). For each balloon profile all MIPAS profiles within a radius of 1000 km and a temporal offset of 24 hours (48 hours for 19 September 2003) were taken into account and averaged, resulting in adding up of 10–14 profiles, respectively. The dotted lines indicate the standard deviation of the MIPAS profiles and the error bars (red) the 1-σ measurement precision of the balloon profiles.



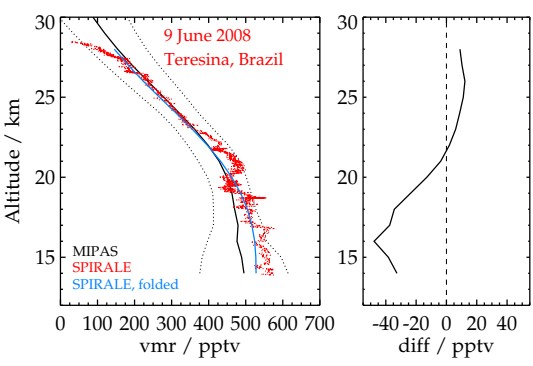
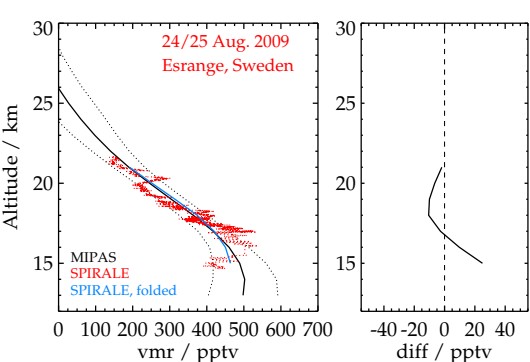

**Figure 3.** Comparison of MIPAS OCS data (black) with OCS balloon profiles obtained by the SPIRALE experiment (red) on 9 June 2008 over Teresina, Brazil (5.1°S, 42.1°W, left), and on 24/25 August 2009 over Esrange, Sweden (67.9°N, 21.1°E, right). The displayed MIPAS data are averages, for which all profiles within a radius of 1000 km and a temporal offset of 24 hours with respect to the SPIRALE profiles were taken into account. The dotted lines indicate the standard deviation of the MIPAS profiles, and the blue curves are the SPIRALE profiles convolved with MIPAS averaging kernels.



**Figure 4.** Comparison of OCS data obtained by MIPAS (solid black curves) and by ACE-FTS (v3.5) (red curves) in the latitude bands 60°– 90° (top row), 30°–60° (middle row) and 0°–30° (bottom row), both for the southern (left) and northern hemisphere (right). The displayed profiles are averages of collocated data in the respective latitude bands. For each ACE-FTS profile, MIPAS data within a maximum temporal offset of 5 hours and a maximum spatial distance of 500 km were taken into account. Only ACE-FTS OCS data with quality_flag = 0 were used. Dashed black curves are the differences between MIPAS and ACE-FTS.




**Figure 5.** Time series of monthly and zonally averaged OCS measured by MIPAS at 30 km, 26 km, 22 km, 18 km, 14 km and 10 km (top to bottom). White areas extending over the whole latitude range are data gaps due to operational shutdown of MIPAS, white areas at 10 km in the equatorial region are caused by cloud contamination (or by scan pattern). Note the changes in VMR-scales.





**Figure 6.** Mean OCS profiles measured by MIPAS in the latitude bands 0°N–30°N, 30°N–60°N, 60°N–90°N, 0°S–30°S, 30°S–60°S and 60°S–90°S (top left to bottom right) during different seasons. Colour coding of seasons: March–May (black), June–August (red), September–November (blue), December–February (green).





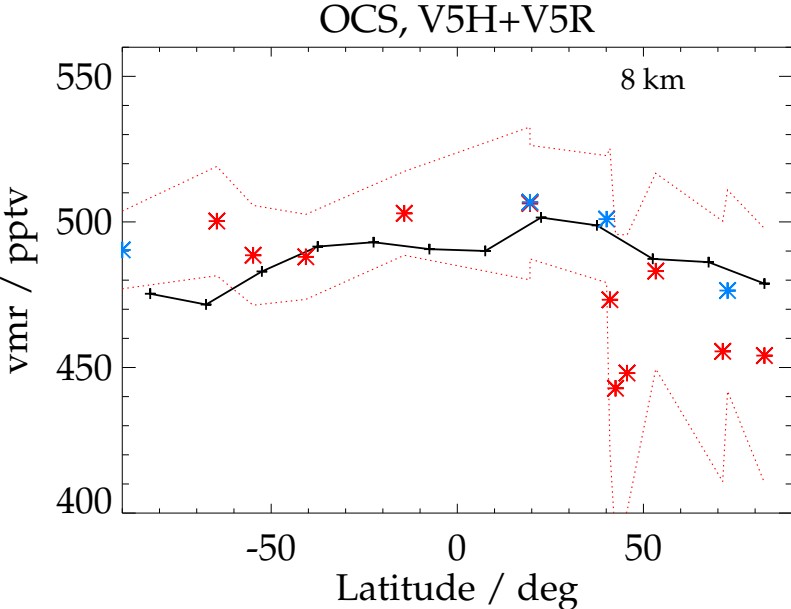

**Figure 7.** Latitudinal variation of MIPAS OCS data at 8 km altitude, averaged over the whole measurement period from 2002 to 2012 (black curve), and OCS amounts (flask data) from NOAA/ESRL surface stations (stars), averaged over the same time period. Surface stations indicated by red stars are situated in the boundary layer and surface stations indicated by blue stars in the free troposphere. The red dotted curves represent the 1-sigma standard deviation of the flask data.






**Figure 8.** Left: OCS latitude-height cross sections measured by MIPAS between 2002 and 2012 during December to February, March to May, June to August and September to November (top to bottom). Middle: Same as left but for EMAC OCS model results. Right: Same as left, but for EMAC OCS model results convolved with a MIPAS averaging kernel of a tropical OCS profile.





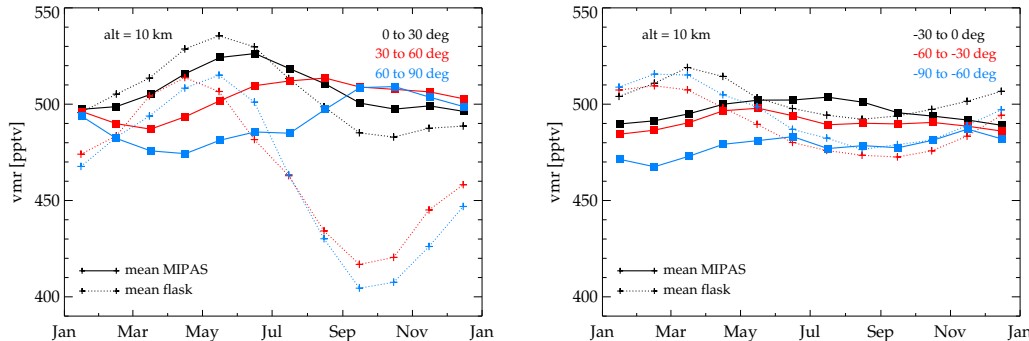

**Figure 9.** Monthly variation of OCS amounts measured by MIPAS during 2002–2012 (solid lines) and at surface stations (dotted lines) in the latitude bands 0°–30° (black), 30°–60° (red) and 60°–90° (blue) in the northern (left) and southern hemisphere (right). Surface data are averaged over all ESRL stations in the respective latitude bands and over the whole measurement period of MIPAS.

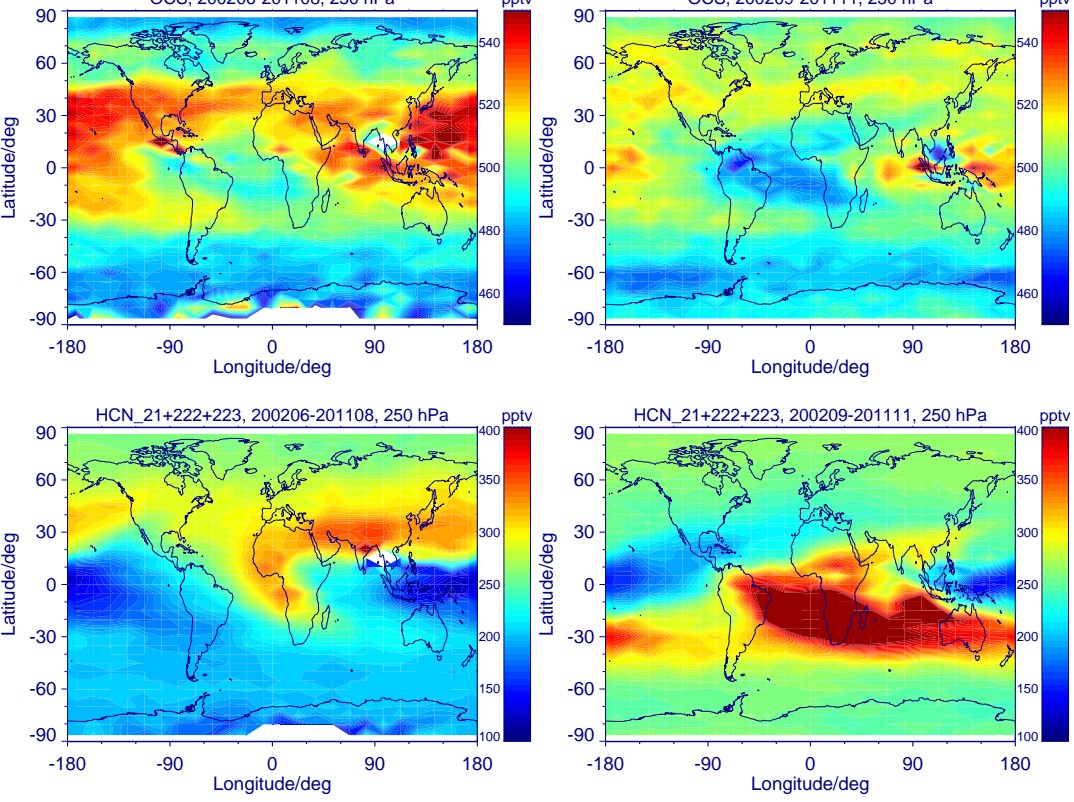

**Figure 10.** Top row: OCS distribution measured by MIPAS between 2002 and 2012 at the 250 hPa level during the periods June to August (left) and September to November (right). Bottom row: Same as top panel, but for MIPAS HCN.





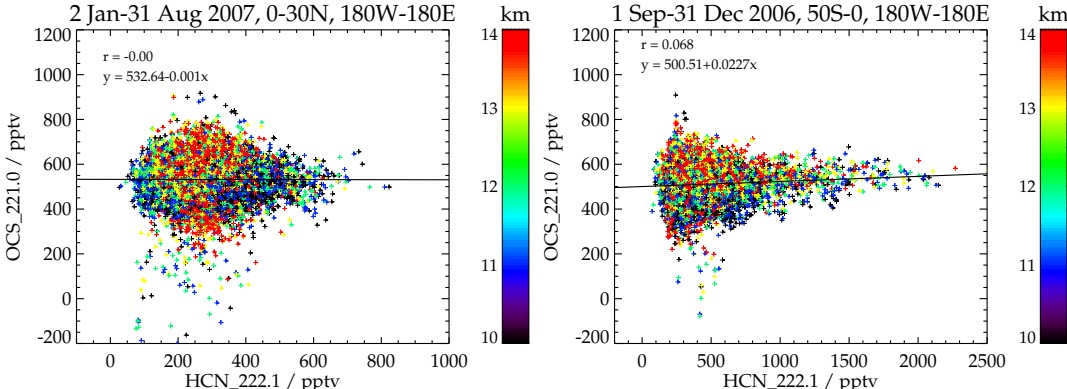

**Figure 11.** MIPAS OCS plotted versus MIPAS HCN observed in the altitude range 10 to 14 km. Left: Measurements between January 2 and August 31, 2007, in the latitude band 0°–30°N. Right: Measurements between September 1 and December 31, 2006, in the latitude band 50°S–0°. The black lines are fitted regression lines, "r" denotes the correlation coefficient and the prefactor of "x" the slope of the regression lines.

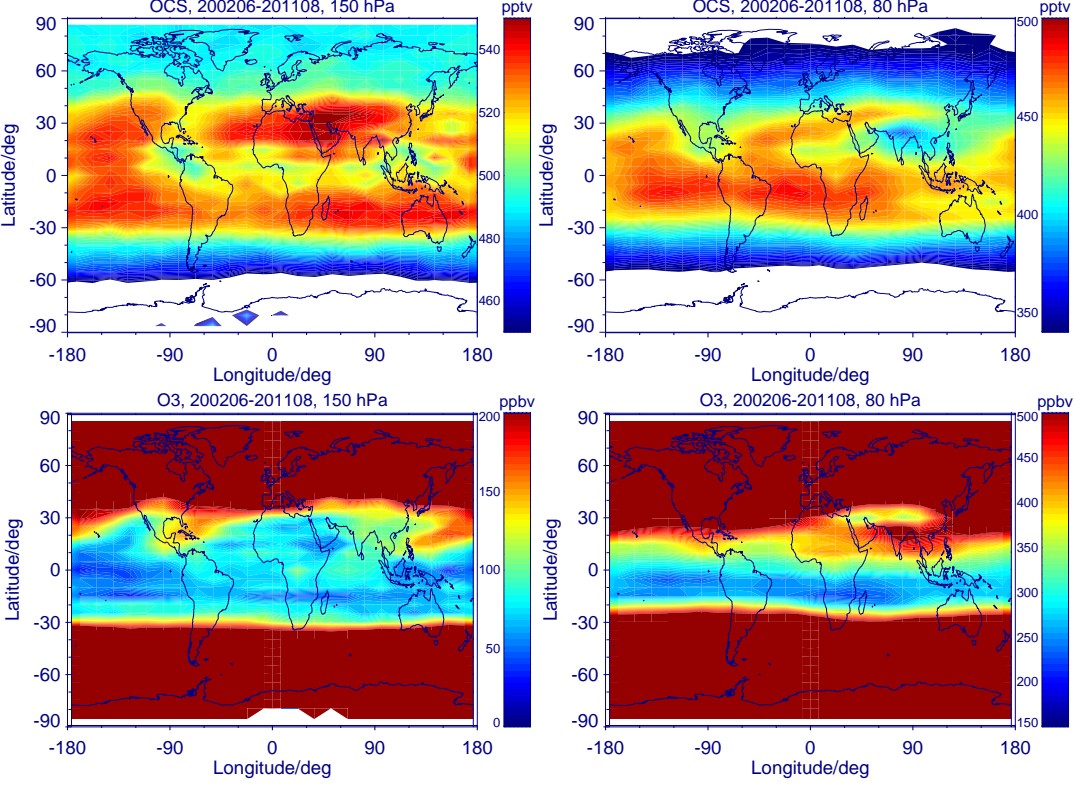

**Figure 12.** Top row: OCS distribution measured by MIPAS between June and August 2002–2012 at the 150 hPa (left) and at the 80 hPa level (right). Bottom row: Same as top row, but for MIPAS $O_3$.





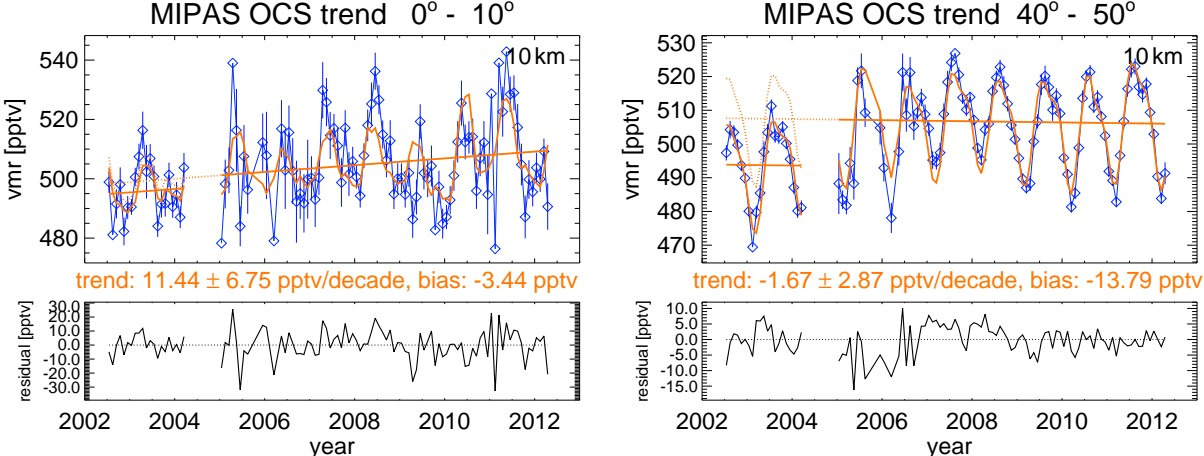

**Figure 13.** Trend analysis of MIPAS OCS measured at 10 km altitude in the latitude bands $0°–10°$N (left) and $40°–50°$N (right). The upper panels show monthly means of measured and fitted OCS (blue and red curves) and fitted linear trends (red lines). The lower panels show the residuals.

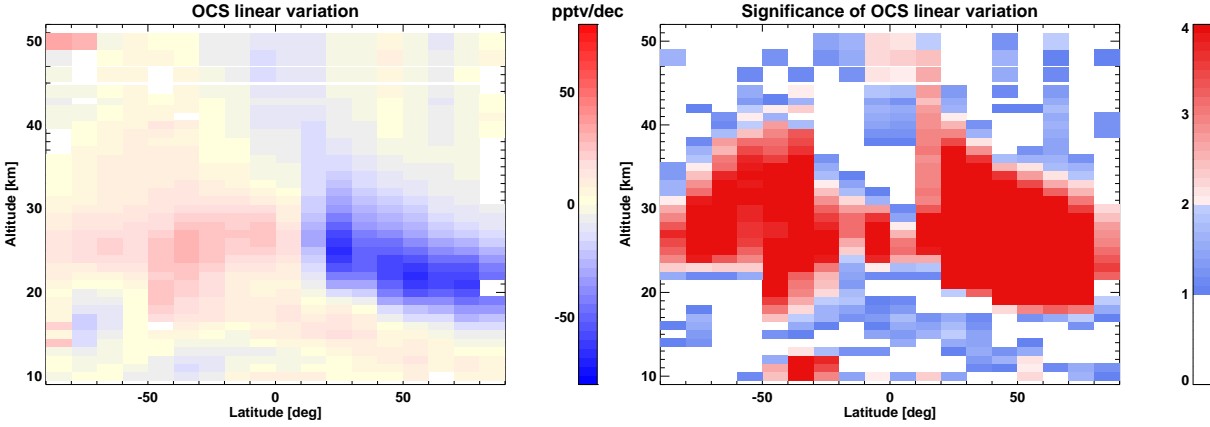

**Figure 14.** Left: Decennial linear trend of monthly averaged OCS volume mixing ratios in $10° \times 1$ km latitude-altitude bins for the period 2002–2012. Right: Significance of the bin-wise trends, estimated by division by the standard deviation from the respective regression analysis.