# Peer review of "Global carbonyl sulfide (OCS) measured by MIPAS/Envisat during 2002–2012"

_Atmospheric Chemistry and Physics, 2016_

## Referee Comment (RC1) · Anonymous Referee #1 · 20 Oct 2016

**1   General Comments**

The subject of the paper is validation and interpretation of a satellite data set of OCS useful for several applications in the fields covered by ACP. There are already several short papers around where this dataset was introduced, but nevertheless, the actual manuscript gives new views and is worth to be published. A very interesting aspect for users are the details on the consequences of averaging kernels used for retrieval. For example it is possible to reproduce "observed" negative mixing ratios with models if the model results are convolved with the averaging kernels.

Please give less focus to the uncertain uppermost levels.

[Figure]

**2 Specific Comments**

In the abstract a sentence on the model simulations is missing.

In the introduction on page 3, line 10 also the recent data by NOAA available on the internet should be mentioned since they are obviously used in Figure 7.

Near the end of section 2.2 it should be explicitly said that the averaging kernels cause a significant high bias in the layers above about 30km.

It might be dangerous to interprete the sparse data in the upper troposphere in the tropics in detail because of possible biases due to longitudinal features like for example more clouds above the West Pacific warmpool (sections 4.1 and 4.2.2., first paragraphs). Please add more information here.

The Montzka data cited at the beginning of section 4.2.1 do not represent a zonal mean in Northern midlatitudes, but are low biased because most stations are located near the Eastcoast, where the influence of the uptake by vegetation and soil is strongest because of the prevailing winds.

Please cite the comparison of the OCS data with EMAC in a different setup in Brühl et al (2015, J. Geophys. Res. Atmos. 120, 2103) in section 4.2.2 (third paragraph). Does the version in the manuscript contain a flux boundary condition for OCS? Please reformulate paragraph 4 of section 4.2.2: There is no stronger upwelling in the MIPAS data compared to the model, this is just an artifact introduced by the averaging kernels as demonstrated by the third column of Figure 8. Here it is nice to see that the convolution with the model results even reproduces the negatives. This figure is a very valuable contribution for modelers using satellite data.

For comparison with MIPAS is would be more useful to keep the mountain stations and give the 3 Eastcoast stations a lower weight by just using their average. These numbers should be given in addition in the text of section 4.2.3.

The seasonal cycle of OCS observed by MIPAS in Figure 9 (section 4.2.4) in high and midlatitudes is influenced by the polar vortices which should be mentioned. In the Southern hemisphere this effect might by reduced by a sampling artifact in MIPAS data because vortex airmasses have PSCs and therefore data gaps. It would be useful to add two frames in Figure 9 with corresponding model results to disentangle dynamical and surface effects.

Section 4.3.2 should be shortened. It is not necessary to use 2 figures and a table to demonstrate that there is no correlation, at least Figure 11 is superfluous in the main text (skip and replace by a sentence or move to a supplement).

Please improve text at the beginning of section 4.3.3. Isn't that tongue due to the Northern part of the AMA circulation?

The discontinuity at the time of the switch between the data versions should be addressed in the text of section 4.4 or the caption of Figure 13. A continuous line would be not in contradiction to the Montzka data and better fit the Jungfraujoch data. Does the analysis in Figure 14 also include the jump in 2004? Figure 14 points to a kind of redistribution or an oscillation of circulation patterns. Please skip the sentences on the insignificant and uncertain upper layers.

**3  Technical Corrections**

The ticks on the time axis in Figure 5 should be thicker and longer.

Include "i.e. on mountains" in caption of Figure 7.

---

## Referee Comment (RC2) · Anonymous Referee #2 · 28 Oct 2016

Review of Glatthor et al. "Global carbonyl sulfide (OCS) measured by MIPAS/Envisat during 2002–2012"

This paper presents an overview of the upper tropospheric/stratospheric carbonyl sulfide (OCS) between 5–40 km observed by the satellite instrument MIPAS during 2002–2012. The same dataset has been already used in a previous paper [Glatthor et al., 2015, doi: 10.1002/2015GL066293]. In this manuscript, the authors first compare the vertical OCS profiles observed by MIPAS against balloon profiles, ACE-FTS and ground-based measurements. Then the authors describe in details the spatial distribution and the temporal variation OCS, such as annual variations and interhemispheric differences. The authors also discuss the contributions of tropospheric processes such as biomass burning and surface sources/sinks to the observed variability of OCS in the upper tropospheric region.

The very detailed documentation of the spatiotemporal variability of OCS is attractive. But a rigorous statistical treatment of the OCS data, especially in the upper troposphere, is required to justify the results of the current manuscript. Below are the comments that the authors may consider during their revision.

Major comments:

1. There have not been any quantitative analysis of the retrieval characteristics. A figure like Figure 3 in Millán et al. [2015, ACP, doi:10.5194/acp-15-2889-2015] is needed to justify the sensitivity of their retrievals to OCS at different altitudes. E.g. How do the sensitivity profiles (i.e. Jacobians) and the averaging kernels look like? What is the a priori concentration? These information have not been presented in Glatthor et al. [2015] nor the current manuscript. However, these information are critical if the authors want to discuss the OCS variability in the upper troposphere or below, as to show how much of the retrieved values actually come from the a priori and the measurement.

2. The retrieval error analysis is not complete. In Section 2.2, only an estimate of the total errors in the troposphere (50 ppt) and the stratosphere (120 ppt) are presented. The authors explain that the measurement noise is the dominant error. I assume what has been taken as "total errors" in this manuscript is the sum of the "random error" and the forward model parameter errors, defined in von Clarmann et al. [2003]'s Eqs (2) and (3) respectively. von Clarmann et al's Eqs (2) and (3) are the same as Eqs (3.30) and (3.18) of Rodgers [2000] respectively, and do not include the smoothing error (Eq. 3.29 of Rodgers [2000]), which is partly due to the deviation from the a priori and partly due to the Twomey-Tikhonov constraint. Indeed, the smoothing error was not discussed in von Clarmann et al. [2003]. In the current manuscript, the authors used a height-independent constant profile as the a priori. But the OCS concentration varies strongly with height across the tropopause. Therefore, the smoothing error due to the deviation from the a priori should depend on the vertically constant concentration they have assumed. Furthermore, the authors should also mention the error due to ambient temperature.

3. In addition to Comment #2, the 1000 micron band used in this work and Glatthor et al. [2015] for the OCS retrieval is 100 times weaker than the strongest OCS absorption band at 2040 micron that has been used by the IASI and TES teams. The OCS absorption signals at some

selected altitudes (e.g. 7 km, 10 km, 20 km, etc) should be compared to the instrument noise, in a similar way in Figure 1 of Millán et al. [2015], to illustrate that the OCS signal is strong enough for retrieval purpose.

4. The fact that the seasonal patterns are obtained with a constant a priori is quite promising but the authors should also plot the evolution of the error terms in the same way as in their Figure 5 to show that the seasonal patterns are not results of errors.

5. The comparison between MIPAS OCS and other OCS measurements are not consistent. The SPIRALE data have been convolved with the averaging kernel before comparing MIPAS whereas ACE-FTS and MkIV have not. The authors explain that it was because the vertical resolution of SPIRALE is higher. However, in additional to the degradation of the vertical resolution, the effect of the a priori in the MIPAS OCS is also applied to the SPIRALE through the averaging kernel. Therefore, the averaging kernel is applied either to all datasets or to none.

Minor comments:

1. Page 1, Line 17: Should "tropospheric OCS" be actually "upper tropospheric OCS"? The authors mostly discuss the OCS in the upper troposphere near 10 km or 250 hPa. But in the abstract (and in the text), the authors sometimes refer to "tropospheric OCS" (e.g. for the trends). The authors should clarify whether they are actually referring the upper tropospheric OCS or really tropospheric OCS, say, in the model simulations or inferred from HCN or ozone data.

2. Page 2, Line 23-24: "A comprehensive compilation of these budget estimations is given in Kremser et al. (2016)." Do you mean Kremser et al. (2015)?

3. Page 3, Line 4-5: How would Lejeune et al. [2011] discuss the trend in 2012? Please check.

4. Page 3, Line 24: Somehow the authors should also mention IASI and TES OCS products for completeness because this manuscript discusses the tropospheric OCS.

5. Page 5, Line3: What's values of a priori used by retrieval? What did you use for constraint of a priori? Could you show the profile of a priori with uncertainties you used in Figure 1?

6. Page 5, Line 15: 41–48 pptv and 10–26% cannot be both right.

7. Figure 2: Is the same a priori profile used at all locations? If not, it may be better to show the a priori profile in each panel.

8. Page 8, Line 3: Has there been any explanation why there was an increase of OCS concentration at 14 km after 2006?

9. Page 11, Line 25: The term "slower convection" is contradictory. Should it be "vertical mixing" or "upwelling"?

10. Page 13, Line 29-31, 'In the northern hemisphere there is a band of enhanced values

extending from the tropical Atlantic to the Chinese coast, which reflects the upper end of the Asian Monsoon Anticyclone including westward outflow.' Could this high OCS extending from Atlantic to China also result from Arabic anticyclone?

11. The discussion of Asian Monsoon anticyclone (AMA) signature of OCS distribution at UTLS is illuminating on the underlying transport. The authors may want to explain more clearly what mechanisms caused the pattern of enhanced OCS on the north end and low OCS on the top of AMA.

---

## Author Comment (AC1) · 23 Dec 2016

[12pt,a4paper]article

[dvips]color

**Response to reviewer 1:**

We thank reviewer 1 for her/his helpful comments. Please find below our responses describing how the manuscript has been modified with respect to the comments. Blue passages denote changes or updates in the revised manuscript.

**1. General Comments**

[Figure]

*"Please give less focus to the uncertain uppermost levels."*

Reply: Actually we do not give much focus to the uppermost levels. In the discussion of Figures 5 and 8 there are only a few sentences, in which we list some general stratospheric features visible in OCS: the QBO, upwelling in the tropics and subsidence in the polar vortices. Further, we trace back the difference between measured and modelled upwelling to the MIPAS averaging kernels. We think these stratospheric observations are robust and can be maintained.

Change: The discussion of the insignificant trends at 40 km altitude in the last two sentences of Section 4.4 has been removed (also on demand of reviewer 2).

**2. Specific Comments**

Comment: *"In the abstract a sentence on the model simulations is missing."*

Reply: The sentence "Simulations with the ECHAM-MESSy model reproduce the observed latitudinal cross sections fairly well." has been added.

Comment: *"In the introduction on page 3, line 10 also the recent data by NOAA available on the internet should be mentioned since they are obviously used in Figure 7."*

Reply: Thank you for pointing this out. We added the sentence "NOAA/ESRL data are available at http://www.esrl.noaa.gov/gmd/hats/gases/OCS.html." in paragraph

5 of the introduction. Further, we added an acknowledgement for provision of the NOAA/ESRL/GMD flask data.

Comment: *"Near the end of section 2.2 it should be explicitly said that the averaging kernels cause a significant high bias in the layers above about 30km."*

Reply: As suggested by reviewer 2 a plot of the averaging kernels has been added to Figure 1. In the discussion of this plot we now state that "In the stratosphere the AKs are centered at increasingly lower altitudes with a displacement of up to 2 km at 40 km altitude. This indicates that the OCS signal is actually from somewhat further below and that symmetrical averaging kernels would lead to lower OCS values."

Comment: *"It might be dangerous to interpret the sparse data in the upper troposphere in the tropics in detail because of possible biases due to longitudinal features like for example more clouds above the West Pacific warmpool (sections 4.1 and 4.2.2., first paragraphs). Please add more information here."*

Reply: To account for the reviewer's concerns, we added the sentence "Measurements at 10 km altitude above Indonesia and the western Pacific warmpool are nearly permanently impeded by clouds, while cloud contamination in the remaining tropical latitude band exhibits a moderate sesonal variation." However, in our discussion of tropical upper tropospheric data in sections 4.1 and 4.2.2 there are no conclusions on strong interannual OCS differences, which might be biased by varying cloud coverage.

Comment: *"The Montzka data cited at the beginning of section 4.2.1 do not represent a zonal mean in Northern midlatitudes, but are low biased because most stations are located near the Eastcoast, where the influence of the uptake by vegetation and soil is*

*strongest because of the prevailing winds."*

Reply: The reviewer is right, but there has already been an explanation for this low bias and the discrepancy to upper tropospheric zonal means in section 4.2.1 of the discussion paper. To make things clearer, we changed the wording into "The largest deviations occur at two ground-based stations at northern mid-latitudes, where the OCS amounts are lower by 50 pptv. These stations are located in the central (LEF, 45.9°N, 90.3 °W) and eastern United States (HFM, 42.5°N, 72.2 °W), where the OCS amounts are strongly reduced by vegetative uptake during the growing season (Montzka et al., 2007)."

Comment: *"Please cite the comparison of the OCS data with EMAC in a different setup in Brühl et al (2015, J. Geophys. Res. Atmos. 120, 2103) in section 4.2.2 (third paragraph)."*

Reply: The sentence "A comparison of MIPAS OCS with EMAC simulations from a different setup has been presented in Brühl et al. (2015)." has been added in section 4.2.2 (third paragraph).

Comment: *"Does the version in the manuscript contain a flux boundary condition for OCS?"*

Reply: Yes, it does. To point this out the sentence "In the EMAC simulations presented here, monthly varying OCS emissions taken from the scenario of Kettle et al. (2002) and modified by enhancement of the tropical vegetation uptake, are applied as flux boundary conditions." has been added to paragraph 3 of Section 4.2.2 of the revised manuscript.

[Figure]

Comment: *"Please reformulate paragraph 4 of section 4.2.2: There is no stronger upwelling in the MIPAS data compared to the model, this is just an artifact introduced by the averaging kernels as demonstrated by the third column of Figure 8. Here it is nice to see that the convolution with the model results even reproduces the negatives. This figure is a very valuable contribution for modelers using satellite data."*

Reply: The fact that the apparent stronger upwelling in the MIPAS data is an artefact caused by the averaging kernels has already been outlined in the discussion paper. To make things even clearer we changed the passage "generally exhibit a stronger upwelling" into "apparently exhibit a stronger upwelling" and rephrased the subsequent sentences as follows:

"However, convolution of the EMAC data with the averaging kernels of a MIPAS OCS profile obtained in the tropics (right column) leads to much better agreement in the tropical upper stratosphere and to somewhat better agreement in the northern hemispheric troposphere and in the transition region between 15 and 25 km altitude. Thus the apparent stronger upwelling of measured OCS in the tropical upper stratosphere is an artefact caused by the displaced averaging kernels in this region (cf. Section 2.2)."

Comment: *"For comparison with MIPAS is would be more useful to keep the mountain stations and give the 3 Eastcoast stations a lower weight by just using their average. These numbers should be given in addition in the text of section 4.2.3."*

Reply: Actually, as taken from Montzka et al. (2007), the northern hemispheric ratio has alread been given even for complete omission of the Midwest and Eastcoast station (1.0) at the end of Section 4.2.3.

Additional point: after having re-read the Montzka-reference, we noticed that our statement "In this estimation all northern hemispheric sites situated at more than 3000 m above sea level had been excluded." was wrong and replaced it by "In this estimation the results from the stations THD, MHD and SUM had been excluded, because they did not cover the whole measurement period.".

Comment: *"The seasonal cycle of OCS observed by MIPAS in Figure 9 (section 4.2.4) in high and midlatitudes is influenced by the polar vortices which should be mentioned. In the Southern hemisphere this effect might by reduced by a sampling artifact in MIPAS data because vortex airmasses have PSCs and therefore data gaps. It would be useful to add two frames in Figure 9 with corresponding model results to disentangle dynamical and surface effects."*

Reply: We added the sentence "Of course, other processes, as e.g. subsidence of OCS-poor air masses in the polar vortices, can also contribute to the differences between the variations at the ground stations and observed by MIPAS at higher northern latitudes. Nevertheless, ..." We decided not to show model results because they do not help to explain the differences between the ground stations and MIPAS observations.

Comment: *"Section 4.3.2 should be shortened. It is not necessary to use 2 figures and a table to demonstrate that there is no correlation, at least Figure 11 is superfluous in the main text (skip and replace by a sentence or move to a supplement)."*

Reply: We agree. Section 4.3.2 has been shortened by removal of Figure 11 and of the respective discussion on page 12, lines 14–21 in the ACPD-version of the manuscript.

Comment: *"Please improve text at the beginning of section 4.3.3. Isn't that tongue due to the Northern part of the AMA circulation?"*

Reply: We are not quite sure what the reviewer means here. At the beginning of section 4.3.3 we do not talk about a "tongue", but particularly discuss enhanced OCS amounts at 150 hPa in the AMA. We talk about a tongue of enhanced ozone in section 4.3.4, which like the region of depleted OCS is attributed to meridional transport of extra-tropical air around the AMA.

Comment: *"The discontinuity at the time of the switch between the data versions should be addressed in the text of section 4.4 or the caption of Figure 13. A continuous line would be not in contradiction to the Montzka data and better fit the Jungfraujoch data."*

Reply: We adressed the discontinuity by adding the sentence "As outlined in Section 4.1 there is a discontinuity at or somewhat after the switch from the HR to the RR mode. This is taken into account by an offset as additional fit parameter, resulting in biases of 5 and 14 pptv in the two examples, respectively."

Fitting of a continuous line over both measurement periods would probably lead to moderate positive trends in the upper troposphere, which would be in agreement with the Jungfraujoch data between 2002 and 2008, but in contradiction to the Jungfraujoch data after 2008 and also to the Montzka data. Further, we found discontinuities between HR and RR mode data for several other MIPAS gases and have performed trend analyses for a couple of species by taking into account an offset between both periods (see e.g. Kellmann et al. (2012) or Eckert et al. (2014) in the reference list).

Comment: *"Does the analysis in Figure 14 also include the jump in 2004? Figure 14 points to a kind of redistribution or an oscillation of circulation patterns. Please skip the sentences on the insignificant and uncertain upper layers."*

Reply: The analysis in Figure 14 contains both measurement periods, i.e. also the jump in 2004. Yes, as outlined in our manuscript we also assume that Figure 14 points to a change - most probably a southward shift - of the Brewer-Dobson circulation. The last two sentences of Section 4.4 on the less significant trends in the tropical stratosphere above 40 km altitude and at high southern latitudes between 14 and 20 km have been skipped.

Comment: *"The ticks on the time axis in Figure 5 should be thicker and longer."*

Reply: Figure 5 was modified accordingly.

Comment: *"Include "i.e. on mountains" in caption of Figure 7."*

Reply: We changed the respective sentence into "Surface stations indicated by red stars are situated at low altitudes in the boundary layer and surface stations indicated by blue stars on mountains in the free troposphere."

---

## Author Comment (AC2) · 23 Dec 2016

[12pt,a4paper]article

[dvips]color

**Response to reviewer 2:**

Many thanks for reading our manuscript and your helpful comments. Please find below our responses describing how the manuscript has been modified with respect to your annotations. Blue passages denote changes or updates in the revised manuscript.

**Comment:**

*"The very detailed documentation of the spatiotemporal variability of OCS is attractive. But a rigorous statistical treatment of the OCS data, especially in the upper troposphere, is required to justify the results of the current manuscript."*

We reply to this below in the context of the major comments.

**Major comments:**

*"1. There have not been any quantitative analysis of the retrieval characteristics. A figure like Figure 3 in Millan et al. [2015, ACP, doi:10.5194/acp-15-2889-2015] is needed to justify the sensitivity of their retrievals to OCS at different altitudes. E.g. How do the sensitivity profiles (i.e. Jacobians) and the averaging kernels look like? What is the a priori concentration? These information have not been presented in Glatthor et al. [2015] nor the current manuscript. However, these information are critical if the authors want to discuss the OCS variability in the upper troposphere or below, as to show how much of the retrieved values actually come from the a priori and the measurement."*

Reply: Thank you for pointing this out. We added a plot with MIPAS OCS averaging kernels to Figure 1, which we discuss in the last paragraph of Section 2.2. We do not use optimal estimation but a Tikhonov first order regularization, and the a priori concentration is zero at all altitudes. Our OCS data do not depend on this particular number, because our regularization does not push the result to the a priori but only smooths the retrieval. Any other altitude-constant a priori would lead to the same results, because our regularization constrains only the shape of the profiles but not the values.

[Figure]

*"2. The retrieval error analysis is not complete. In Section 2.2, only an estimate of the total errors in the troposphere (50 ppt) and the stratosphere (120 ppt) are presented. The authors explain that the measurement noise is the dominant error. I assume what has been taken as "total errors" in this manuscript is the sum of the "random error" and the forward model parameter errors, defined in von Clarmann et al. [2003]'s Eqs (2) and (3) respectively. von Clarmann et al's Eqs (2) and (3) are the same as Eqs (3.30) and (3.18) of Rodgers [2000] respectively, and do not include the smoothing error (Eq. 3.29 of Rodgers [2000]), which is partly due to the deviation from the a priori and partly due to the Twomey-Tikhonov constraint. Indeed, the smoothing error was not discussed in von Clarmann et al. [2003]. In the current manuscript, the authors used a height-independent constant profile as the a priori. But the OCS concentration varies strongly with height across the tropopause. Therefore, the smoothing error due to the deviation from the a priori should depend on the vertically constant concentration they have assumed. Furthermore, the authors should also mention the error due to ambient temperature."*

Reply: In Section 2.2 we indeed give only two estimates of the total error in the troposphere and in the stratosphere. But we refer to Table 1, where the total errors and several error components are given with a better height resolution. We did not want to repeat the content of the whole table in the text. The meaning of the total error has already been given in the captions of Figure 1 and now has also been added to the last but one sentence of Section 2.2. The smoothing error is not contained in our error analysis, because we chose to characterise our retrieval setup by the averaging kernels instead. Even Rodgers does not claim that error estimation is only complete with the smoothing error included. In Rodgers (2000, Sect. 3.2.1) it reads "For the purpose of carrying out an error analysis, the retrieval can be either regarded as an estimate of a state smoothed by the averaging kernel rather than an estimate

of the true state, or as an estimate of the true state, but with an error contribution due to smoothing". We have chosen the first alternative for reasons discussed in von Clarmann (2014). The error due to the temperature uncertainty is very small and also given in Table 1.

*"3. In addition to Comment 2, the 1000 micron band used in this work and Glatthor et al. [2015] for the OCS retrieval is 100 times weaker than the strongest OCS absorption band at 2040 micron that has been used by the IASI and TES teams. The OCS absorption signals at some selected altitudes (e.g. 7 km, 10 km, 20 km, etc) should be compared to the instrument noise, in a similar way in Figure 1 of Millan et al. [2015], to illustrate that the OCS signal is strong enough for retrieval purpose. "*

Reply: We assume the reviewer refers to the bands at 860 and 2040 cm$^{-1}$. Thank you for pointing this out, but please note that both IASI and TES are nadir sounders. For limb emission sounding of the upper troposphere the band at 860 cm$^{-1}$ is not weaker but actually about 10 times stronger than the strongest OCS absorption band at 2040 cm$^{-1}$. In addition to the radiances spectral noise has also to be taken into account, which for MIPAS spectra is about 10 times higher in the 860 cm$^{-1}$ region. Thus, both bands seem to be equally well suited. However, since the spectral signatures in the 2040 cm$^{-1}$ region become saturated for upper tropospheric observations, we decided to use the band at 860 cm$^{-1}$. We added a new Figure 1 containing the OCS signatures at 10 and 20 km along with spectral noise to the manuscript, which illustrates that the band is strong enough for retrieval purpose. This figure is discussed in paragraph 2 of Section 2.2.

*"4. The fact that the seasonal patterns are obtained with a constant a priori is quite promising but the authors should also plot the evolution of the error terms in the same way as in their Figure 5 to show that the seasonal patterns are not results of errors."*

Reply: Due to the heavy computational load required, a full error analysis was performed for selected MIPAS scans only. The only error estimate available for every MIPAS scan is the estimated standard deviation (ESD). Thus, we plotted the evolution of the ESD of the bin-averaged OCS values in the same way as in Figure 5. These plots generally show lower variations and no correlation with the seasonal pattern of the volume mixing ratios (see attached Figure).

*"5. The comparison between MIPAS OCS and other OCS measurements are not consistent. The SPIRALE data have been convolved with the averaging kernel before comparing MIPAS whereas ACE-FTS and MkIV have not. The authors explain that it was because the vertical resolution of SPIRALE is higher. However, in additional to the degradation of the vertical resolution, the effect of the a priori in the MIPAS OCS is also applied to the SPIRALE through the averaging kernel. Therefore, the averaging kernel is applied either to all datasets or to none."*

Reply: The constraint applied to MIPAS measurements by the IMK processor is a first order Tikhonov constraint. Contrary to usual optimal estimation retrievals, where the a priori covariance matrix is used as a regularization matrix, the absolute OCS amount is in the null space of the regularization matrix. Thus, the constraint can only reduce the altitude resolution but cannot push the retrieval towards the a priori. However, since MIPAS Aks become vertically displaced in the stratosphere, they have also been applied to the MkIV and ACE-FTS profiles, and the discussion in Sections 3.1 and 3.3 has been slightly adjusted.

**Minor comments:**

*"1. Page 1, Line 17: Should "tropospheric OCS" be actually "upper tropospheric OCS"? The authors mostly discuss the OCS in the upper troposphere near 10 km or 250 hPa. But in the abstract (and in the text), the authors sometimes refer to "tropospheric OCS" (e.g. for the trends). The authors should clarify whether they are actually referring the upper tropospheric OCS or really tropospheric OCS, say, in the model simulations or inferred from HCN or ozone data."*

Reply: We changed the wording into "upper tropospheric OCS" on page 1, lines 17 and 20.

*"2. Page 2, Line 23-24: "A comprehensive compilation of these budget estimations is given in Kremser et al. (2016)." Do you mean Kremser et al. (2015)?"*

Reply: No, we mean the review paper of Kremser et al. (2016), as given in the reference list.

*"3. Page 3, Line 4-5: How would Lejeune et al. [2011] discuss the trend in 2012? Please check."*

Reply: The correct year should have been "2010". However, very recently an updated analysis of the Jungfraujoch trends until the year 2015 has been presented in Lejeune et al. (2017). We now cite the trends given in this publication.

Comment:*"4. Page 3, Line 24: Somehow the authors should also mention IASI and TES OCS products for completeness because this manuscript discusses the tropospheric OCS."*

[Figure]

Reply: We agree. To account for IASI and TES OCS products, the sentence "Space-borne OCS measurements of the NASA Aura Tropospheric Emission Spectropmeter (TES) and of the Infrared Atmospheric Sounding Interferometer (IASI) have been presented by Kuai et al. (2014) and by Vincent and Dudhia (2016), respectively." has been added at the end of paragraph 5 of the Introduction.

*"5. Page 5, Line 3: What's values of a priori used by retrieval? What did you use for constraint of a priori? Could you show the profile of a priori with uncertainties you used in Figure 1?"*

Reply: We used a zero a priori profile and added this information "(zero at all altitudes)" to the manuscript. As already indicated in the preceding sentence of the manuscript, a first order Tikhonov smoothing constraint was applied. The effect of the a priori and the constraint is fully characteriszed by the averaging kernels. We do not think it is necessary to show the zero a priori profile applied.

*"6. Page 5, Line 15: 41–48 pptv and 10–26% cannot be both right."*

Reply: These values are both right, because the corresponding retrieved OCS mixing ratios are 414.0 pptv (15 km) and 182.5 pptv (20 km).

*"7. Figure 2: Is the same a priori profile used at all locations? If not, it may be better to show the a priori profile in each panel."*

Reply: The same height-constant a priori profile (zero at all altitudes) is used for each

[Figure]

MIPAS geolocation.

*"8. Page 8, Line 3: Has there been any explanation why there was an increase of OCS concentration at 14 km after 2006?"*

As outlined in Section 4.1 of the manuscript, the main increase of OCS at 14 km occurs in the second half of 2005 and in our opinion is due to the change of the observation mode. As correctly observed by the referee, there is a slight additional increase in 2006. We changed the wording in the updated manuscript into "Another slight increase occurs in 2006, just as at 18 km altitude. The reason for this increase, either geophysical or instrumental, is unclear."

*"9. Page 11, Line 25: The term "slower convection" is contradictory. Should it be "vertical mixing" or "upwelling"?"*

Reply: We changed "slower convection" into "weaker vertical mixing".

*"10. Page 13, Line 29-31, "In the northern hemisphere there is a band of enhanced values extending from the tropical Atlantic to the Chinese coast, which reflects the upper end of the Asian Monsoon Anticyclone including westward outflow." Could this high OCS extending from Atlantic to China also result from Arabic anticyclone?"*

Reply: As far as we know, there is no "Arabic anticyclone" in contrast to the "Asian monsoon anticyclone". Instead, there are two modes of the Asian monsoon anti-cyclone, the "Iranian mode" and the "Tibetan mode". During one monsoon season the anticyclone moves once or several times between the modes, occasionally also

splitting up, followed by westward or eastward outflow. Thus, we think the feature in Fig. 12, which is averaged over the summer months of 9 years, reflects the Asian Monsoon Anticyclone.

*"11. The discussion of Asian Monsoon anticyclone (AMA) signature of OCS distribution at UTLS is illuminating on the underlying transport. The authors may want to explain more clearly what mechanisms caused the pattern of enhanced OCS on the north end and low OCS on the top of AMA."*

Reply: We do not quite understand, what the referee means here. Is it about differences between 150 and 80 hPa? In this case our interpretation of the plots is that at 150 hPa the AMA extends over a wide subtropical area and that at 80 hPa just the central part of the AMA is still visible.

**References:**

von Clarmann, T.: Smoothing error pitfalls, Atmos. Meas. Tech., 7, 3023-3034, doi:10.5194/amt-7-3023-2014, 2014.

---

## Author Comment (AC3) · 23 Dec 2016

Time series of the ESD of the bin-averaged OCS values measured by MIPAS at 30 km, 26 km, 22 km, 18 km, 14 km and 10 km (top to bottom). The white area in 2004 is a data gap due to operational shutdown of MIPAS and the remaining white areas are caused by cloud contamination. Note the different colour ranges. Values outside of the colour ranges are also labelled in black or red, respectively.